# The dynamic proteome of influenza A virus infection identifies M segment splicing as a host range determinant

Boris Bogdanow[1,2,7], Xi Wang [1,8], Katrin Eichelbaum[1], Anne Sadewasser [2], Immanuel Husic[1], Katharina Paki[2], Matthias Budt[2], Martha Hergeselle[1], Barbara Vetter[3], Jingyi Hou[1], Wei Chen [1,4], Lüder Wiebusch [3], Irmtraud M. Meyer[1,5], Thorsten Wolff [2] & Matthias Selbach [1,6]*

Pandemic influenza A virus (IAV) outbreaks occur when strains from animal reservoirs acquire the ability to infect and spread among humans. The molecular basis of this species barrier is incompletely understood. Here we combine metabolic pulse labeling and quantitative proteomics to monitor protein synthesis upon infection of human cells with a human- and a bird-adapted IAV strain and observe striking differences in viral protein synthesis. Most importantly, the matrix protein M1 is inefficiently produced by the bird-adapted strain. We show that impaired production of M1 from bird-adapted strains is caused by increased splicing of the M segment RNA to alternative isoforms. Strain-specific M segment splicing is controlled by the 3′ splice site and functionally important for permissive infection. *In silico* and biochemical evidence shows that avian-adapted M segments have evolved different conserved RNA structure features than human-adapted sequences. Thus, we identify M segment RNA splicing as a viral host range determinant.

[1] Max Delbrück Center for Molecular Medicine, Robert-Rössle-Strasse 10, 13125 Berlin, Germany. [2] Unit 17 "Influenza and other Respiratory Viruses", Robert Koch Institut, Seestrase 10, 13353 Berlin, Germany. [3] Labor für Pädiatrische Molekularbiologie, Charité Universitätsmedizin Berlin, Augustenburger Platz 1, 13353 Berlin, Germany. [4] Department of Biology, Southern University of Science and Technology, Xuanyuan Road 1088, 518055 Shenzhen, China. [5] Freie Universität Berlin, Department of Biology, Chemistry, Pharmacy Institute of Chemistry and Biochemistry, Thielallee 63, 14195 Berlin, Germany. [6] Charité Universitätsmedizin Berlin, 10117 Berlin, Germany. [7] Present address: Structural Interactomics, Leibniz-Forschungsinstitut für Molekulare Pharmakologie, Robert-Rössle-Strasse 10, 13125 Berlin, Germany. [8] Present address: Division of Theoretical Systems Biology, German Cancer Research Center, 69120 Heidelberg, Germany. *email: matthias.selbach@mdc-berlin.de

nfluenza A viruses (IAVs) are negative-sense, single-stranded RNA viruses with a segmented genome. IAV infection causes seasonal epidemics and sporadically pandemic outbreaks in the human population with significant morbidity, mortality, and economic burden. IAVs can infect both mammals (e.g., humans, pigs, horses) and birds (e.g., chicken, waterfowl). However, strains that are replicating in birds typically do not infect mammals and vice versa. Pandemics occur when influenza strains of avian origin with novel antigenicity acquire the ability to transmit among humans[1]. Understanding the molecular basis of host specificity is therefore of high medical relevance.

The species barriers that hinder most avian IAVs from successfully infecting humans are effective at several steps in the viral life cycle. For example, the avian virus receptor hemagglutinin (HA) recognizes oligosaccharides containing terminal sialic acid (SA) that are linked to galactose by α2,3[2]. In the human upper respiratory airway epithelium, the dominant linkage is of α2,6 type, to which human-adapted HA binds. Despite these differences in receptor binding, many avian viruses are internalized by human cells and initiate expression of the viral genome. Such infections typically lead to an abortive, nonproductive outcome in mammalian cell lines. Our understanding of this intracellular restriction is still incomplete. One well-established factor is the influenza RNA-dependent RNA polymerase (RdRP): this enzyme catalyzes replication of the viral genome and transcription of viral messenger RNAs (mRNAs)[3]. Polymerases from avian strains are considerably less active in mammalian cells than their counterparts from mammalian-adapted strains[4]. A wealth of experimental data described adaptive mutations that alter receptor specificity or fusion activity of HA (reviewed in ref. [5]) and polymerase activity (reviewed in ref. [6]). However, relatively little is known about the contribution of other IAV genes for permissive vs. non-permissive infection[7].

A crucial aspect for permissive infection is the correct timing of viral gene expression: IAV proteins are produced at the specific phase of infection when they are needed[8]. One example is the M gene, which encodes predominantly two polypeptides: the larger protein, M1, is produced from a collinear transcript. The smaller one, M2, is encoded by a differentially spliced transcript[9]. M1 is the matrix protein with multiple functions that encapsulates the viral genome, and also mediates nuclear export[10,11]. M2 is a proton-selective channel that is an integral part of the viral envelope[12,13]. The ratio of spliced to unspliced products increases during infection[14], which reflects the changing demands required for optimal viral replication.

Systems-level approaches have provided important insights into the molecular details of host–virus interaction[15]. For example, RNA interference (RNAi) screens identified host factors required for IAV replication[16–18]. Also, interaction proteomics experiments identified many cellular binding partners of IAV proteins[19–21]. A number of studies also quantified changes in protein abundance[22–27]. However, these steady-state measurements cannot reveal the dynamic changes in protein synthesis during different phases of infection. Early studies used radioactive pulse labeling to monitor protein synthesis in IAV-infected cells[28,29]. However, radioactive pulse labeling cannot provide kinetic profiles for individual proteins. More recently, stable isotope labeling by amino acids in cell culture (SILAC) emerged as a powerful means to study the dynamic proteome[30]. SILAC-based pulse labeling methods such as pulse SILAC (pSILAC) and dynamic SILAC can quantify protein synthesis and degradation on a proteome-wide scale[31,32]. Moreover, metabolic incorporation of bioorthogonal amino acids such as azidohomoalanine (AHA) provides a means to biochemically enrich for newly synthesized proteins[33]. In combination with SILAC, AHA labeling can be used to quantify proteome dynamics with high temporal resolution[34–36].

Here, we use metabolic pulse labeling and quantitative mass spectrometry to compare proteome dynamics upon infection of human cells with a human-adapted and a bird-adapted IAV strain. We find that host proteins behave surprisingly similar, but observe striking differences in the production of viral proteins, especially for the matrix protein M1. Follow-up experiments with reporter constructs, in silico studies, and reverse genetics identify an evolutionarily conserved cis-regulatory element in the M segment as a host range determinant.

## Results

**Quantifying the dynamic proteome of IAV infection**. To assess species specificity of IAVs, we used a model system comparing a low-pathogenic avian H3N2 IAV (A/Mallard/439/2004—Mal) to a seasonal human IAV isolate of the same subtype (A/Panama/2007/1999—Pan). While the avian virus is not adapted to efficient growth in cultured human cells and causes a non-permissive infection, the seasonal human virus replicates efficiently. We demonstrated previously that the Pan virus produces >1000-fold more infectious viral progeny than the non-adapted virus, even though both strains efficiently enter human cells and initiate their gene expression program[27].

We reasoned that comparing the kinetics of protein synthesis upon infection with both strains might reveal determinants of species specificity. To this end, we performed proteome-wide comparative pulse-labeling experiments by combining labeling with AHA and SILAC (Fig. 1a): cells incorporate AHA instead of methionine into newly synthesized proteins when the cell culture medium is supplemented with this bioorthogonal amino acid. AHA contains an azido group, which can be used to covalently couple AHA-containing proteins to alkyne beads via click chemistry. In this manner, newly synthesized proteins can be selectively enriched from the total cellular proteome. Combining AHA labeling with SILAC reveals the kinetics of protein synthesis with high temporal resolution[36]. First, we fully labeled human lung adenocarcinoma cells (A549) using SILAC. Second, individual cell populations were infected with either Pan or Mal virus or left uninfected. Third, all cells were pulse labeled with AHA for 4 h during different time intervals post infection (0–4, 4–8, 8–12, and 12–16 h). The three cell populations for every time interval were then lysed, combined, and AHA-containing proteins were enriched from the mixed lysate using click chemistry (Fig. 1b). After on-bead digestion, peptide samples were analyzed by high-resolution shotgun proteomics.

We quantified proteins using two readouts: (i) SILAC-based relative quantification to assess differences in de novo protein synthesis and (ii) intensity-based absolute quantification (iBAQ) to quantify absolute amounts of newly synthesized proteins[37]. Our data thus provides kinetic profiles for relative and absolute differences in de novo protein synthesis across the course of infection (Supplementary Data 1). In total, we identified 7729 host and 10 viral proteins and quantified 6029 proteins in at least two biological replicates with overall good reproducibility (Supplementary Fig. 1).

**The dynamic host proteome**. It is well established that IAV induces a global reduction in the production of host proteins. This host shutoff was attributed to a plethora of viral effector functions[38]. To assess the host shutoff in our proteomic data, we investigated iBAQ values for viral and host proteins. As expected, viral proteins were potently induced, while the production of host proteins decreased over time (Fig. 1c). The difference between host and viral protein synthesis reached several orders of magnitude and was highest during the 8–12 h pulse interval. Moreover, the total cellular protein output dropped to ~24% (Pan) or

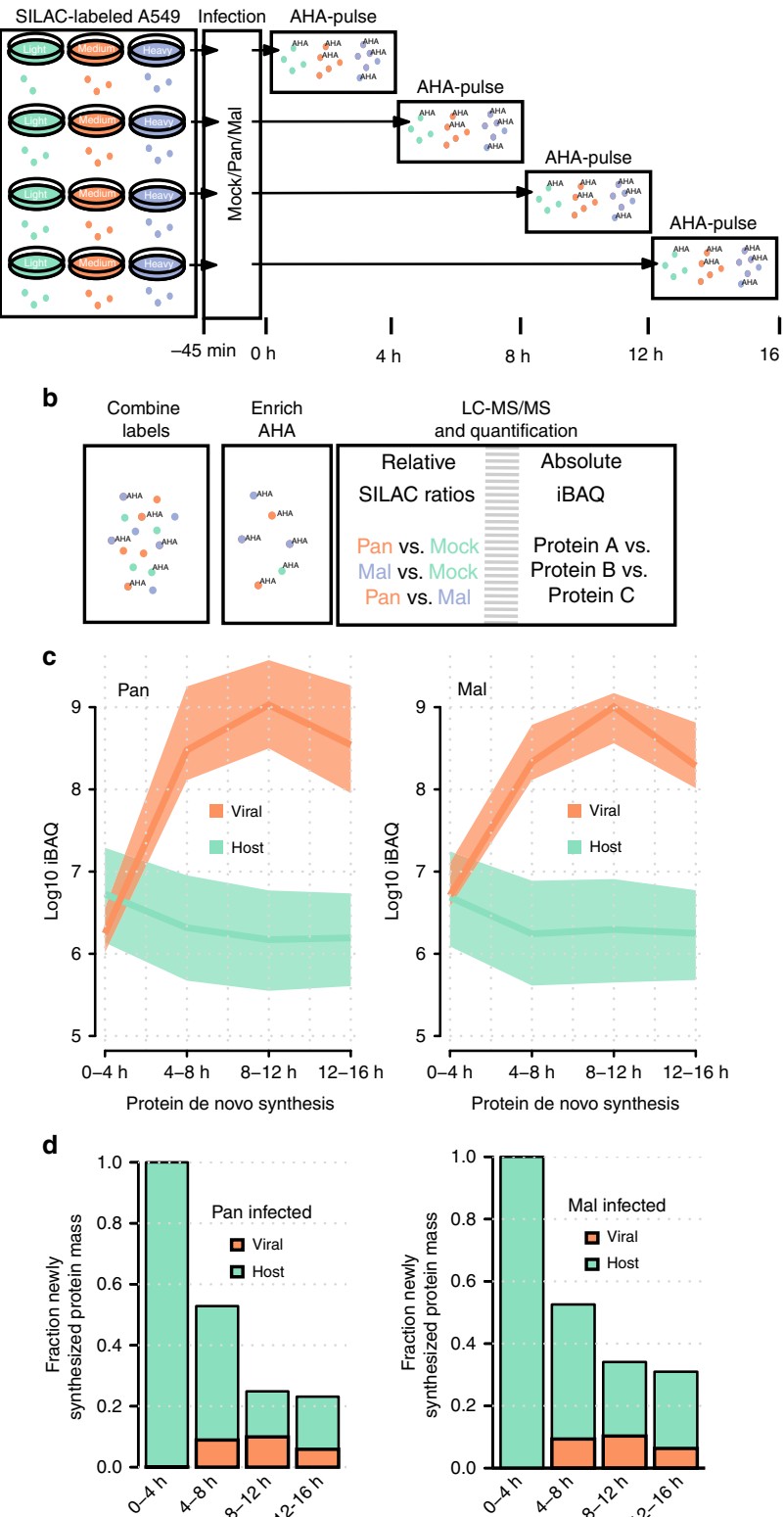

**Fig. 1 A strategy to quantify protein de novo synthesis proteome-wide. a** SILAC-L/M/H-labeled A549 cells were infected with the human seasonal H3N2 IAV isolate Panama (Pan) or the avian H3N2 isolate Mallard (Mal) or left uninfected. The methionine analogous AHA was given to the methionine-depleted medium in different 4 h intervals. **b** After lysis and enrichment for proteins that incorporated AHA, samples were subjected to shotgun proteomics. Absolute and relative protein synthesis profiles were quantified for host and viral proteins. **c** iBAQ-based quantification of protein synthesis levels for host and viral proteins in cells infected with either Pan or Mal virus, as indicated. Median, 25th and 75th percentile of the respective populations are given. **d** Quantification of the total newly synthesized protein mass for host and viral proteins with either Pan or Mal infection as indicated. Data were normalized to the 0–4 h time period. All data are based on the mean of $n = 2$ biological replicates.

~33% (Mal) at later stages of infection, of which ~20–40% was of viral origin (Fig. 1d). At this level of detail, we observed no major differences between both strains. Thus, both strains initiate viral protein synthesis and induce the shutoff of host protein synthesis to an overall similar extent.

Next, we investigated the profiles of individual host proteins across the course of infection. For this, we directly looked at SILAC ratios comparing infected and non-infected cells (Fig. 2a, b). As expected, synthesis of the vast majority of host proteins markedly decreased over time. However, some proteins were less affected by the host shutoff and displayed only a mildly decreased or even increased production. To assess this observation more systematically, we selected the proteins that were least affected by the shutoff at different pulse periods and performed gene ontology (GO) analysis. The heatmap of enriched GO terms provides a global overview of biological processes as the infection progresses (Fig. 2c and Supplementary Data 2). For example, many well-known interferon-induced antiviral defense proteins (e.g., MX1, several IFIT proteins, several oligoadenylate synthase proteins) were relatively strongly produced at late stages of infection. Also, many ribosomal proteins (GO term "peptide chain elongation") largely escaped the host shutoff. Interestingly, we also observed significant enrichment of proteins involved in steroid metabolism and mitochondrial proteins (mito-ribosomal, respiratory chain proteins) at early and intermediate stages of infection, respectively. Cellular responses to infection with the Pan and Mal strain were overall similar. To assess potential differences between permissive and non-permissive infection, we compared protein log 2 fold changes between both viruses directly (Fig. 2d). Interestingly, type I interferon response proteins were first preferentially produced during non-permissive infection. At later stages, however, infection with the Pan virus elicited a stronger interferon response.

Several different hypotheses were made to explain the IAV-induced host shutoff. This included mechanisms at the transcriptional[39], post-transcriptional[40,41], and translational[42] level. To study the relationship of mRNA and protein levels, we quantified mRNA levels at 8 h post infection by RNA-sequencing (RNA-seq). mRNA level differences at this time point showed good correlation with corresponding differences in de novo protein synthesis, particularly during the subsequent 8–12 h period (Fig. 2e–g). Thus, mRNA level changes play an important role for the shutoff of individual mRNAs, corroborating the view that the host shutoff is mainly due to reduced host mRNA levels[41]. The correlation was higher for the Mal than for the Pan strain, suggesting that infection with the latter strain involves additional post-transcriptional processes.

Traditionally, IAV is thought to prioritize the translation of viral over host mRNAs[42], but more recent experimental and computational analyses challenge this view[41,43]. We investigated this question by calculating protein synthesis efficiencies (i.e., the amount of protein made per mRNA). To this end, we divided iBAQ values by corresponding RPKM (Reads Per Kilobase Million) values (Fig. 2h). Infection with both strains reduced host protein synthesis efficiencies compared to uninfected controls. Importantly, we did not observe preferential translation of viral transcripts. This suggests that mRNAs from human and avian influenza virus strains access the translational machinery with comparable efficiency, which argues against the idea that modulation of translation efficiency affects species specificity.

**Dysregulated synthesis of viral proteins**. Since the observed differences in host protein synthesis were surprisingly subtle, we focused our attention to the dynamics of viral protein synthesis. Production of most viral proteins peaked in the 8–12 h period

(see Supplementary Fig. 2). The kinetics such as the early production of NS1 and NP and delayed synthesis of M1 is consistent with classical radioactive pulse labeling experiments[29]. We then used SILAC ratios of shared peptides (i.e., peptides with sequence identity between both strains) to precisely compare the kinetics of viral protein synthesis (Fig. 3a, b). We found that the avian strain produced higher amounts of all viral proteins at the beginning, confirming that the Mal virus successfully enters cells and initiates its gene expression program. Later on, during mid to late phases, the human Pan virus produced most proteins more abundantly than the avian strain. Note that NS1 and M2 are excluded in this analysis because no identical peptides were identified.

It is well established that the RdRP from avian-adapted IAV strains is less active in mammalian cells[4,44]. Thus, we would have expected the production of all viral proteins in the bird-adapted strain to be reduced to a similar extent. In contrast, we observed striking differences in the synthesis of individual proteins: HA was more abundantly produced by the avian strain throughout infection. In contrast, neuraminidase (NA) and particularly matrix protein M1 were stronger produced by the human strain at later stages. These differences in the production of individual viral proteins cannot be explained by the global difference in RdRP activity between strains. Thus, the avian strain displays dysregulated protein production relative to its human counterpart.

We focused our attention on the M1 protein since it showed the largest difference between both strains. The protein is highly conserved between Pan and Mal (~96% amino acid identity) and the most abundant protein in virions[45]. Moreover, M1 is known to mediate export of the viral genome across the nuclear membrane—an essential step during permissive infection[10,11]. Thus, accumulation of M1 at late stages of infection is required for the appearance of viral ribonucleoproteins (vRNPs) in the cytoplasm of infected cells. Interestingly, when investigating the subcellular distribution of the viral nucleoprotein (NP) by immunofluorescence microscopy, we observed efficient export during infection with the Pan strain (Fig. 3c). In contrast, NP was inefficiently exported and accumulated in the nucleus upon Mal infection. These microscopy data is also corroborated by the increased interferon response induced by the Pan strain at later stages of infection (Fig. 2d), which is stimulated by cytosolic viral RNA sensors[46]. We conclude that non-permissive infection correlates with reduced M1 production and impaired nuclear export of NP.

**Increased M segment RNA splicing**. We next sought to investigate the mechanism for the impaired M1 production. To this end, we first quantified the levels of viral mRNAs from our RNA-seq data. In total, the avian virus produced ~2/3 of the mRNA of the human strain with the single largest difference observed for M1 (Fig. 4a). The strain-specific differences in M1 mRNA levels were very similar to the observed differences in M1 protein production (Fig. 4b). Hence, the impaired M1 protein production during non-permissive infection can largely be explained by reduced M1 mRNA levels.

M1 is encoded on segment 7 (i.e., the M segment), which is the most conserved segment between Pan and Mal (~89% nucleotide identity). The M1 protein is produced from a collinear transcript that can be alternatively spliced into three additional isoforms, which all use a common 3′ splice site:[47] the M2 mRNA, which encodes the ion channel M2[12], RNA 3, which is not known to encode a peptide, and M4 mRNA, which is proposed to be translated to an isoform of the M2 ion channel in certain strains[48]. We investigated the relative proportion of these

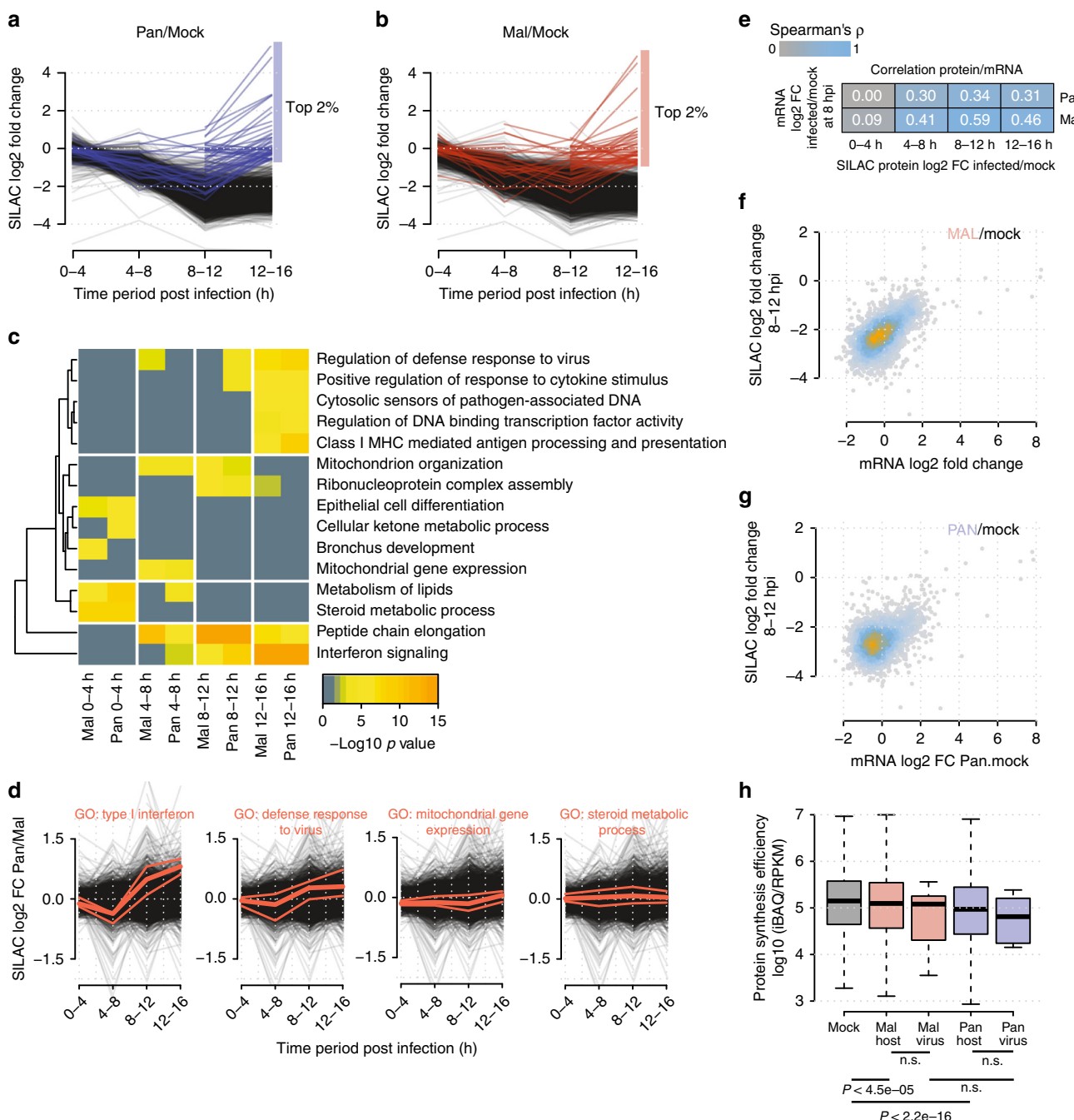

**Fig. 2 The dynamic host proteome of IAV-infected cells. a**, **b** SILAC profiles of host proteins infected with Pan (**a**) or Mal (**b**) compared to mock infection. The top 2% of proteins that are strongest produced during the last time period are highlighted. **c** Functional protein clusters least affected by host shutoff were identified. The 2% proteins with the highest ratios were selected for each time period and multi-list GO enrichment was performed. **d** Median, 25th, and 75th percentile of proteins related to a selected set of GO terms are highlighted in the direct SILAC comparison Pan/Mal. All other protein profiles in black. **e** Spearman's rank correlation coefficients when comparing the indicated RNA and protein level data. **f**, **g** Scatterplot of mRNA changes vs. protein synthesis level changes in Pan/mock or Mal/mock infection conditions. Blue and orange coloring of data points reflects the density of data points. **h** Protein synthesis efficiencies calculated from absolute mRNA (RPKM, 8 hpi) and protein data (iBAQ, 8–12 hpi) (center line, median; box limits, upper and lower quartiles; whiskers, 1.5× interquartile range, outliers (data points outside the whiskers) were removed for visibility). For the assessment of statistical significance two-sided Wilcoxon's rank-sum tests were performed and p values are given (ns: non-significant). All data are based on the mean of $n = 2$ biological replicates.

isoforms in the RNA-seq data via splice junction reads. We detected all known isoforms plus a yet undescribed transcript of the avian M segment, which we call RNA 5. This transcript results from splicing at 5′ donor GG site (pos 520/521) and the common 3′ acceptor site and contains an open reading frame in-frame with M1 with a missing internal region (Fig. 4c).

While only a few percent (2–3%) of the M1 mRNA was alternatively spliced during permissive infection, ~27% was spliced upon infection with the avian strain (Fig. 4d). Thus, the reduced level of M1 mRNA during non-permissive infection is at least partially due to increased splicing of the M1 mRNA to alternative isoforms. In contrast, M2 mRNA levels were rather

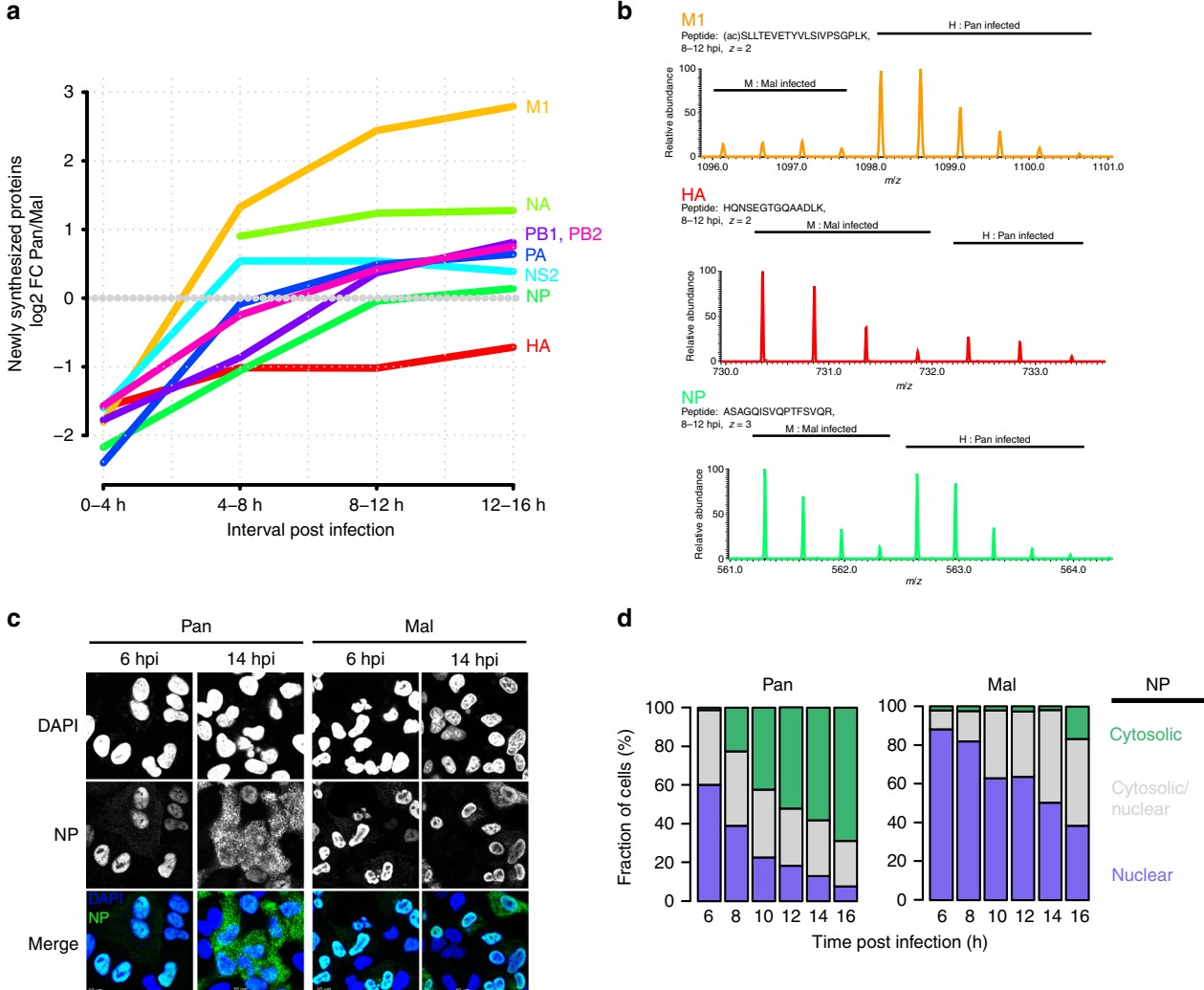

**Fig. 3 Viral protein synthesis is dysregulated during non-permissive infection. a** SILAC protein synthesis profiles of human vs. avian-adapted IAV proteins based on shared peptides. No shared peptides were identified for NS1 and M2. The average of *n* = 2 biological replicates is shown. **b** MS1 spectra of individual precursor (M1—top, HA—middle, NP—lower) peptides in M and H SILAC channels. **c** Cells were infected with the indicated viruses at MOI = 1 (FFU/cell) and analyzed by immunofluorescence microscopy for NP trafficking. Nuclei were counterstained with DAPI. Scale bars: 10 μm. **d** For quantification, NP staining pattern was categorized as predominantly cytosolic, nuclear or both. At least 140 cells were counted per condition.

similar. We note that the comparable M2 mRNA level during non-permissive infection results from two opposing processes— the increased splicing of the primary transcript to the M2 mRNA, and the global reduction in viral transcripts, which is probably due to the impaired polymerase activity[4,44]. We conclude that M segment RNA splicing is markedly different in permissive vs. non-permissive infection.

In principle, the increased splicing of the M segment RNA by bird-adapted strains may reflect an evolutionary adaptation to different needs of the corresponding viral proteins in avian cells. In this case, avian-adapted strains would produce less M1 in both human and avian cells. Alternatively, the optimal balance of viral proteins could be constant and independent of the host species. In this case, avian-adapted strains would produce more M1 in avian than in human cells. To investigate this experimentally, we infected both human and chicken cells with the Mal virus and monitored the production of viral proteins using pSILAC (Fig. 4e). We found that the Mal strain produced considerably more M1 in chicken cells than in human cells (Fig. 4f). Hence, the reduced M1 production of avian viruses in human cells appears to reflect poor adaptation to the mammalian splicing environment.

**A *cis*-regulatory element controls M segment RNA splicing.** The differences in M1 mRNA splicing can be due to (i) *cis*-regulatory elements (i.e., specific signals encoded in the M segment), (ii) *trans*-acting factors (i.e., other viral or host factors that interact with M1 mRNA), or (iii) a combination of both. To assess whether *cis*-regulatory elements are involved, we sought to investigate M segment RNA splicing outside the context of infection. We therefore designed a splicing reporter system (Fig. 5a, b). To this end, we cloned the coding region of the M segment (nucleotides 29–1007) into a eukaryotic expression vector and fused it to an N-terminal Flag/HA tag. Importantly, this construct avoids the strong 5′ splice site of RNA 3[49] and enabled us to assess the relative levels of M1 to M2 proteins. When we transfected human A549 cells with these reporter constructs, we found that M2 was produced to high levels with the construct containing the Mal M sequence, but was barely detectable when the Pan M sequence was transfected (Fig. 5c). Thus, our reporter system recapitulates splicing differences observed during infection. We conclude that *cis*-regulatory elements in the M segment cause excessive splicing of the avian variant.

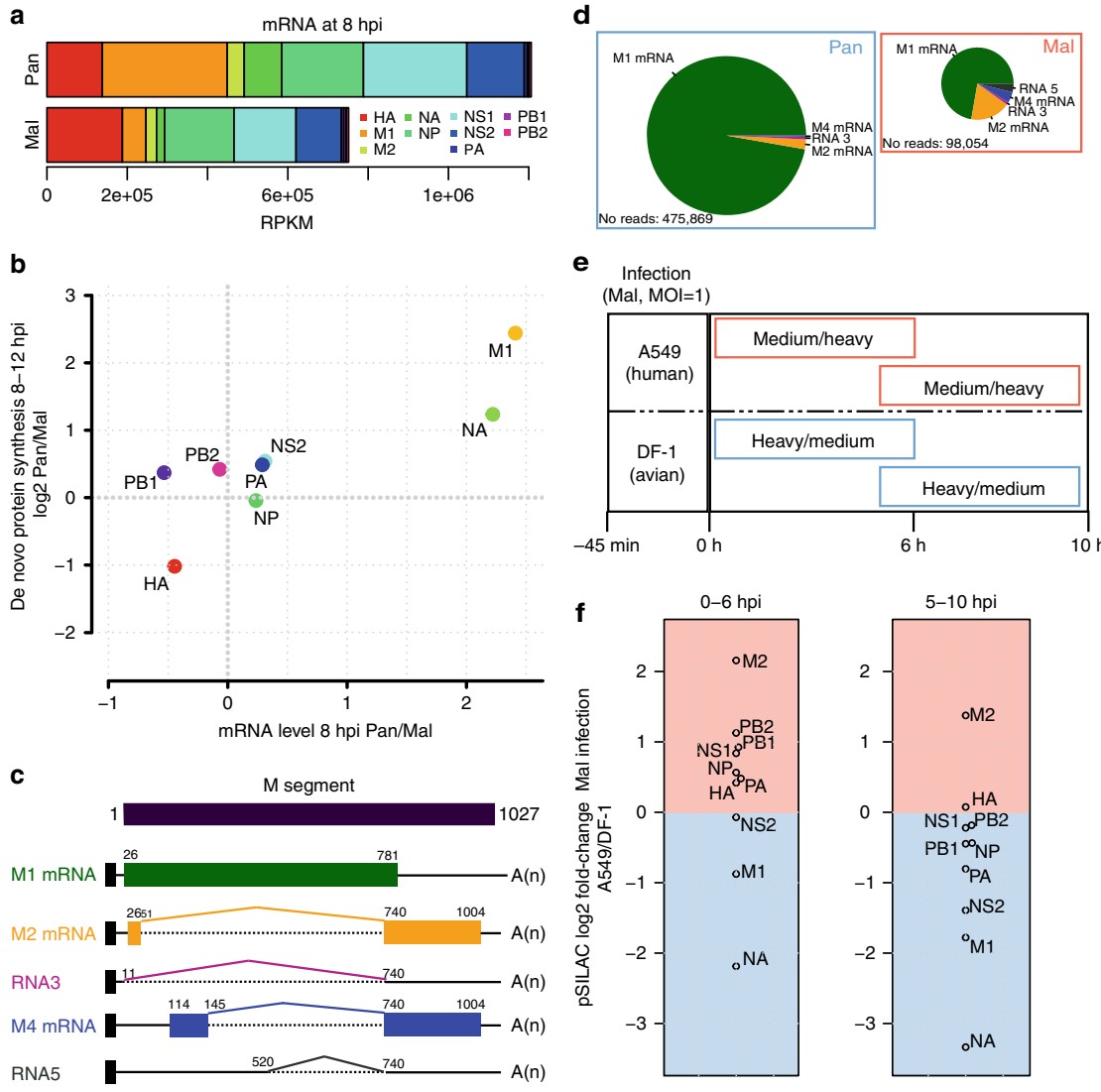

**Fig. 4 Differences in M segment RNA splicing. a** Normalized read counts (RPKM, based on splice junction reads and exonic reads) for 10 viral mRNAs from poly-A-enriched and either Pan- or Mal-infected samples at 8 hpi. The sum of $n = 2$ biological replicates is depicted. **b** Direct comparison of fold changes for viral mRNAs/proteins from RNA level and protein synthesis data. **c** Schematic depiction of the M gene architecture. The M pre-mRNA can be spliced into the indicated isoforms. **d** Relative quantification of the different isoforms based exclusively on splice junction reads from RNA-seq data for both strains at 8 hpi. The area of the pies reflects the absolute number of splice junction reads. The average of $n = 2$ biological replicates is depicted. **e** SILAC-light-labeled DF-1 and A549 cells were pulse labeled with heavy or medium heavy SILAC amino acids (in label-swap duplicates) after infection with the Mal virus (MOI = 1 PFU/cell) and subjected to quantitative shotgun proteomics. **f** pSILAC fold-changes are depicted for both pulse intervals comparing the production of viral proteins with the Mal strain in human vs. avian cells. The average of $n = 2$ biological replicates is depicted.

To determine the sequence responsible for the strain-specific splicing, we made chimeric reporter constructs (Fig. 5b). When swapping the entire intron sequence of the M2 splice variant (nucleotides 52–739, corresponding to ~70% of the coding sequence), we did not observe major changes in the relative amount of M1 to M2. In contrast, integrating the human 3′ splice site region (nucleotides 707–779, 73 nucleotides) into the avian construct strongly impaired splicing down to the levels of the human wild-type construct. Conversely, when we integrated the avian 3′ splice site region into the human construct, we observed a strong increase in splicing, similar to the avian wild-type construct (Fig. 5c). To address splicing in the context of the viral RdRP, we co-transfected HEK293T cells with expression vectors for PB1, PB2, PA, and NP (of WSN strain) and Pan and Mal M wild-type and chimeric segments. In this system, the accumulation of M segment-derived mRNAs depends primarily on the activity of the RdRP. RNA-seq revealed consistent differences in

abundance of the various isoforms with similar overall mRNA levels (Fig. 5d). Thus, the splice site region alone is sufficient to switch the species-specific splicing phenotype in reporter systems using both viral and host RNA polymerases. Interestingly, this region has been reported to contain an RNA secondary structure[50] and a binding site for the splicing factor SRSF1[51]. We conclude that a *cis*-regulatory element in the splice site region determines the strain-specific splicing pattern.

**Evolutionary conserved RNA structure elements in M segments**. We next wanted to assess whether our findings are also relevant for other human- or bird-adapted IAVs. Specifically, we sought to identify functionally relevant RNA secondary structures that have been conserved during evolution of avian- and human-adapted IAVs. To this end, we analyzed multiple sequence alignments (MSAs) from hundreds of recent human H3N2 and avian isolates using the RNA structure prediction program RNA-Decoder[52]

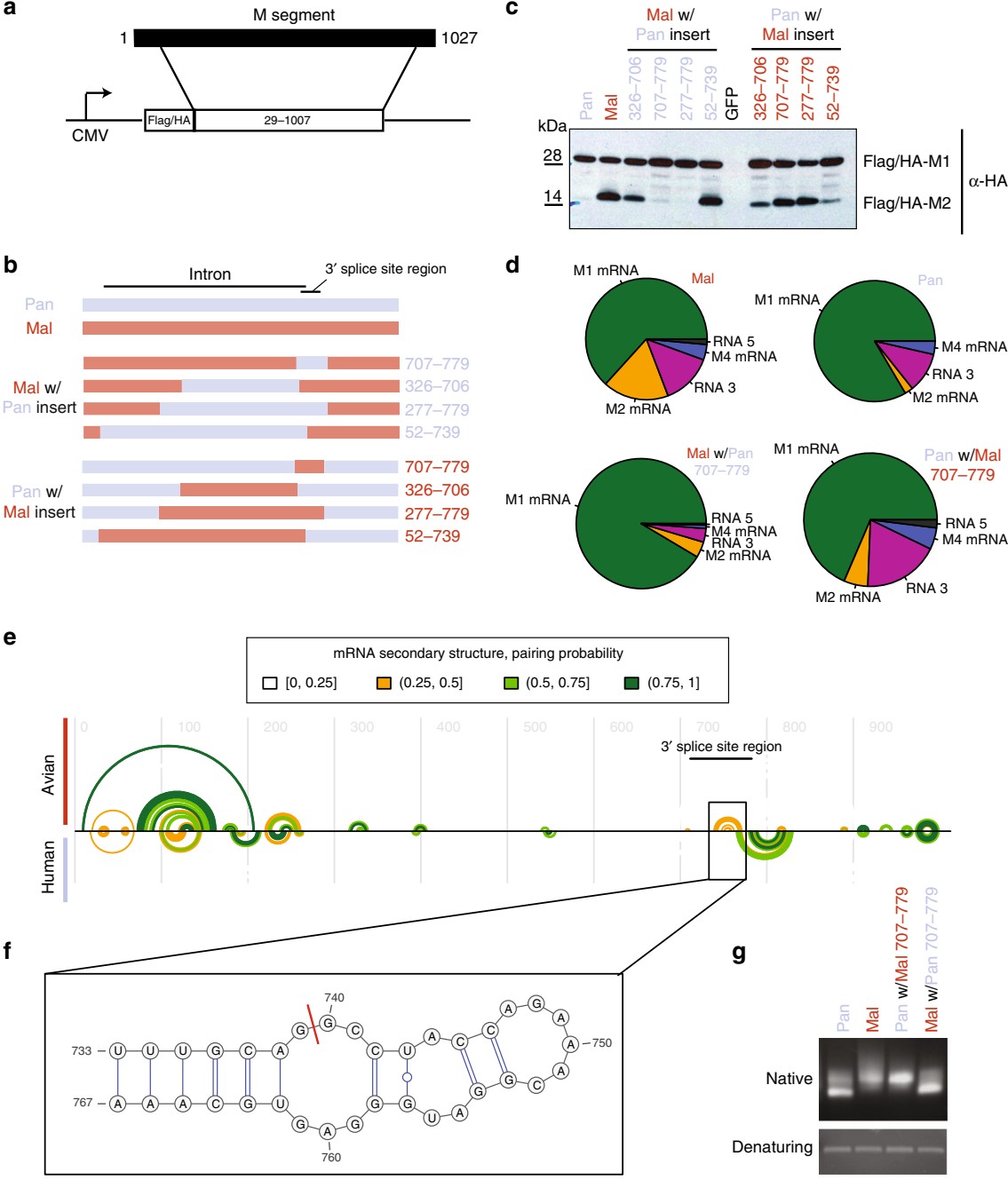

**Fig. 5 A *cis*-regulatory element in the 3′ splice site region controls M segment splicing. a** Reporter system. The coding sequence of segment 7 was cloned into a eukaryotic expression vector with N-terminal Flag/HA tag. **b** Wild-type Pan and Mal sequences as well as several chimeric Pan/Mal constructs were cloned into the reporter vector. **c** A549 cells were transfected with the indicated reporter constructs, harvested, and subjected to anti-HA immunoblotting. **d** HEK293T cells were transfected with constructs encoding NP, PA, PB1, and PB2 (of strain WSN) and vectors encoding for wild-type or chimeric M segments. At 24 h post transfection, RNA was isolated and subjected to RNA-seq. The pies depict the relative counts of isoforms as quantified by splice junction reads normalized to the levels of NP and polymerase subunits. The area of the pies scales with the total number of normalized splice junction reads. The average of $n = 2$ biological replicates is depicted. **e** Evolutionary conserved RNA secondary structure along the M mRNA predicted for avian-adapted (top) and human-adapted H3N2 (bottom) IAV strains. Each semi-circle corresponds to 1 bp involving the corresponding two alignment positions. Color coding of base pairs according to their corresponding, estimated base-pairing probability. The 3′ splice site region (nt 707–779) is annotated with a bar. **f** The insert shows a detail of the predicted RNA secondary structure at the 3′ splice site, predicted for avian but not human sequences in this study. The splice position is indicated by a red bar. **g** The region predicted to be structured with the two species-specific and mutually exclusive elements (707–825) was in vitro transcribed. The RNA was purified and run on native and denaturing agarose gels stained with ethidium bromide.

(see Methods). This program is capable of dis-entangling overlapping evolutionary constraints due to encoded amino acids and RNA structure features and has been shown to successfully identify evolutionarily conserved RNA structures overlapping protein-coding regions, for example, in viral genomes such as hepatitis C and HIV[52,53]. Importantly, RNA-Decoder captures evidence on conserved RNA structure based on the evolutionary signals encoded in the sequences of the input alignment. This is a key

advantage over computational methods that identify RNA structures based on their thermodynamic stability in vitro, as these methods assume that the RNA has no interactions with other molecules (e.g., proteins and other RNAs) in vivo. Also, RNA-Decoder employs a probabilistic framework, which is capable of estimating the reliability of its predictions.

The RNA secondary structure that is best supported by the evolutionary signals in the two MSAs (the so-called maximum-likelihood structure) markedly differs between human and avian strains, particularly in the region around the 3′ splice site (Fig. 5e): the avian region encodes a hairpin-like structure (Fig. 5f) overlapping the 3′ splice site, which is absent from the human-adapted sequences. This structure is similar to a hairpin reported by Moss et al.[50] for four sequences, but differs in details (Supplementary Fig. 3). Most importantly, the evolutionarily conserved structure reported here leaves the GC motif immediately downstream of the AG consensus at the 3′ splice site unpaired, making it potentially more accessible to splicing. In human-adapted sequences, this region is involved in a different conserved secondary structure (Fig. 5e) that is mutually exclusive with this hairpin. We conclude that the M segment of avian and human-adapted isolates contain evolutionarily conserved RNA secondary structures that markedly differ in exactly the region that is critical for strain-specific splicing. Importantly, the consensus sequences of avian and human isolates diverge by five nucleotides in the region of interest (pos: 745, 754, 760, 766, and 772), all of which are retained with the two strains that we investigated here (Supplementary Fig. 4A). Thus, the Pan and Mal strains are by and large representative for human and avian strains in this region.

In addition to these computational analyses, we also wanted to assess RNA secondary structures experimentally. To this end, we in vitro transcribed the region of the M segment containing evolutionarily conserved secondary structure (nucleotides 707–825) from both strains, purified the RNA, and analyzed it on native and denaturing agarose gels (see Fig. 5g). We observed different migration behavior for Pan and Mal under native, but not denaturing conditions, consistent with different secondary structures (lanes 1 and 2). Importantly, when substituting eight nucleotides in the region 707–779 (termed 3′ splice site region in our manuscript) from Mal into the Pan backbone (lane 3), we observed the same migration pattern as for the Mal wild-type RNA. Conversely, when substituting eight nucleotides from the Pan into Mal backbone (lane 4), we observed the same migration pattern as for the Pan wild-type RNA.

To investigate the global significance of these findings, we extended our analyses to other IAV isolates. The M segment of the seasonal Pan strain originates from the M segment of the A/Brevig Mission/1/1918 (p1918) virus, which is at the evolutionary root of human strains and caused the 1918 "Spanish flu" pandemic[54]. Therefore, we cloned the M segment of p1918 into our reporter vector (Supplementary Fig. 4B). Again, we observed inefficient splicing of the p1918 M gene, consistent with our data for the Pan strain and previous reports[55]. Moreover, integration the Mal 3′ splice site region into the p1918 gene increased splicing. Thus, inefficient splicing of the M gene in human-adapted IAVs occurs in a seasonal (Pan) and a pandemic (p1918) strain. The M segment of the Mal strain originates from the eastern avian lineage. A/chicken/Rostock/45/1934 (H7N1), an early representative strain of this lineage, depicted even stronger M segment splicing than the Mal strain (Supplementary Fig. 4C, lane 3). Introducing the human-adapted mutations at the splice site region into this construct completely reversed the splicing phenotype (lane 4). Also, the "avian-like" M segment of A/swine/Netherlands/25/1980 (H1N1) showed a splicing phenotype similar to the Mal strain (lane 2). Finally, the human zoonotic

strain A/Vietnam/1203/2004 (H5N1) exhibited a splicing pattern similar to the Pan strain (lane 5). Collectively, these findings support the global significance of the strain-specific splicing patterns with respect to host range.

**The 3′ splice site region is a host range determinant.** The experiments with reporter constructs described above are advantageous because they allow us to study the impact of M segment sequence features in isolation. Nevertheless, it is also important to assess the relevance of these findings during infection. We therefore mutated eight nucleotides in the splice site region of the Pan wild-type strain to the corresponding nucleotides in the Mal strain using reverse genetics ("Pan-Av" for a Pan strain with an avian splice site region, see Fig. 6a and Supplementary Data 3).

We first compared the kinetics of viral protein synthesis upon infection of A549 cells with both strains using pSILAC[31]. M1 synthesis was selectively impaired during Pan-Av infection during both the 6–12 and 12–18 hours post infection (hpi) time intervals. At later stages, the Pan-Av strain also showed impaired production of other essential viral proteins (Fig. 6b), suggesting that viral replication is also impaired. Next, we quantified M1 and M2 protein (Fig. 6c) and mRNA levels (Fig. 6d, e). The Pan-Av strain displayed decreased M1 protein and mRNA levels, mimicking the behavior of the Mal strain (compare to Fig. 4).

To assess the impact of M segment splicing on IAV replication in human cells, we assessed the growth characteristics of the different viruses (Fig. 6f). As expected, the Pan strain reached ~1000-fold higher titers than the Mal strain. Exchanging the entire M segment of the Pan strain with the M segment of the Mal strain (Pan + Mal M) reduced titers about 10-fold. Importantly, a similar ~10-fold attenuation was also seen in the Pan-Av strain, which only differs from the Pan strain by eight nucleotides. We conclude that the 3′ splice site of the IAV M segment is indeed an important host range determinant.

## Discussion

Advances in high-throughput sequencing have provided insights into the extraordinary diversity of viruses and their genomic determinants of host adaptation. However, the mechanism how these adaptive mutations enable replication in a given host is less understood. Our proteomic pulse labeling data allowed us to take an unbiased look at protein synthesis upon permissive and non-permissive infection. This is advantageous since it allows us to investigate changes in protein synthesis with high temporal resolution, which provides complementary information to more classical steady-state measurements[22–27]. We found that the synthesis profiles of host cell proteins were remarkably similar. Hence, the outcome of infection does not appear to depend on a specific host response. In contrast, we observed striking differences in the synthesis profiles of viral proteins. Particularly, the matrix protein M1 was inefficiently produced during non-permissive infection. Our follow-up experiments showed that this depends— at least partially—on excessive splicing of the avian M1 mRNA to alternative transcripts. Systematic computational analysis of the RNA structure of the M segment revealed characteristic and evolutionarily conserved differences in the splice site regions between human and bird-adapted strains. Exchanging eight nucleotides in the 3′ splice site region from the human-adapted strain to corresponding sequences in the bird-adapted strain markedly impaired replication. Thus, our proteomic analysis of IAV infection identifies M segment splicing as a host range determinant. A hypothetical model for the influence of M segment splicing on the IAV host range is presented in Fig. 7.

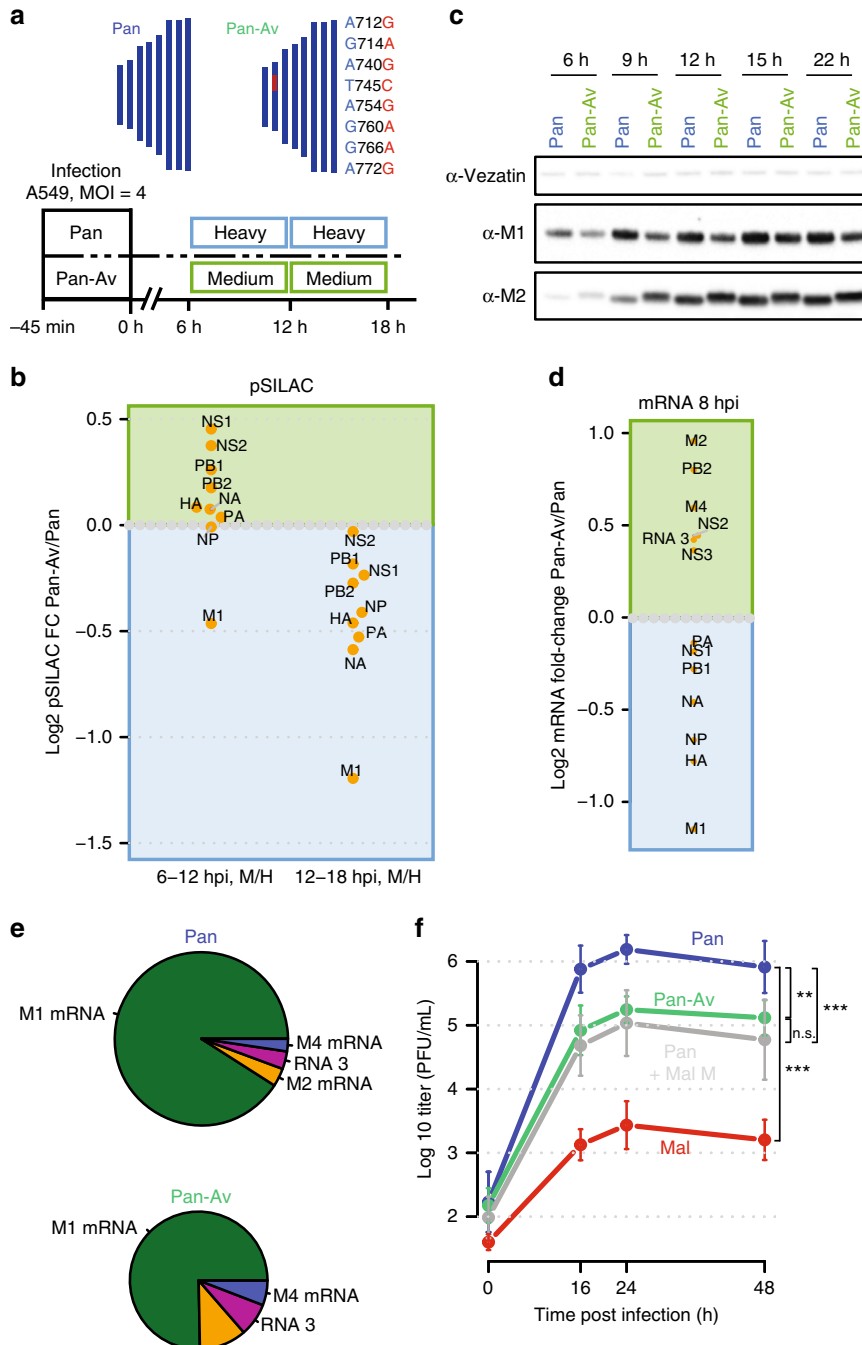

**Fig. 6 The 3′ splice site is a host range determinant. a** (Top) Eight nucleotide polymorphisms of Mal virus at the M segment 3′ splice site were integrated into wild-type Pan virus generating "Pan-Av" mutant. **a** (Bottom) Design of pulsed SILAC experiment. A549 cells were infected with wild-type or mutant virus at an MOI of 4 (FFU/cell) and viral gene expression was assessed by pulse labeling during two time intervals. **b** pSILAC-based quantification of viral protein production comparatively for Pan and Pan-Av virus. One replicate ($n = 1$) is depicted. **c** Kinetics of M1/M2 protein levels as detected by immunoblotting. **d** RNA-seq-based quantification of viral mRNA levels at 8 hpi (MOI = 1 FFU/cell) comparing Pan and Pan-AV infection. The average of $n = 2$ biological replicates is depicted. **e** Quantification of M segment-derived RNAs based on splice junction reads. The area of the pies scales with the total number of splice junction reads. The average of $n = 2$ biological replicates is depicted. **f** Multicycle replication curve of the indicated viruses at an MOI of 0.05 FFU/cell on A549 cells. Means and standard deviations of biological triplicates ($n = 3$) are shown along with significance estimates based on two-sided paired $t$ tests for the 16–48 h time points (n.s.: non-significant, **$p < 0.01$, ***$p < 0.001$).

The cell culture-based infection model and splicing reporter system employed here are advantageous because they enable experiments under well-controlled conditions. Having said this, it is important to keep in mind that these model systems do not represent the full complexity of IAV infections in vivo. Also, the Pan and Mal strain employed here do not represent the full diversity of human- and bird-adapted IAVs. However, our computational analyses show that differences in RNA secondary structure of the 3′ splice site are widely conserved in human- and bird-adapted IAVs. Moreover, recent results from the Steel lab show that avian M segments restrict growth and transmission of mammalian-adapted IAV strains in a guinea pig model[56].

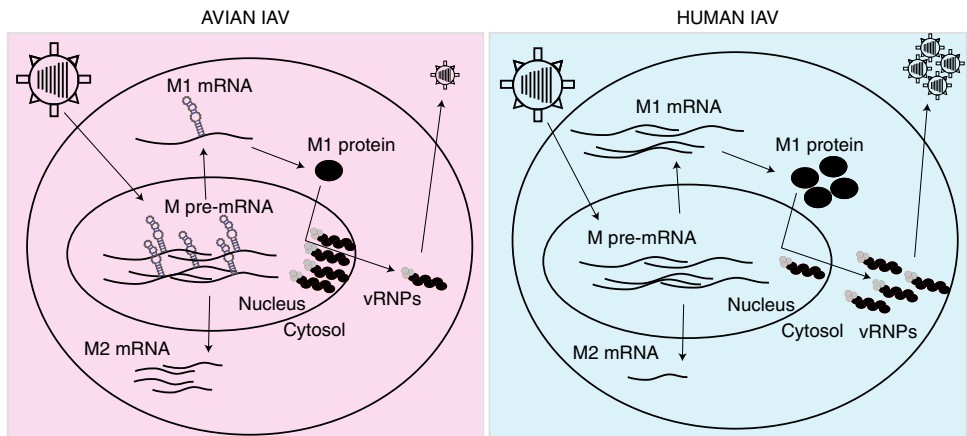

**Fig. 7 Hypothetical model for the role of M segment splicing for IAV host range.** A *cis*-regulatory secondary structure element (indicated hairpins) in avian but not human IAV M pre-mRNA facilitates splicing. This leads to the underproduction of M1 mRNA and protein in human cells infected with avian-adapted IAVs. The poor availability of the M1 protein may contribute to an impaired nuclear export of viral ribonucleoproteins (vRNPs).

Together, these findings strongly suggest that our results are also relevant outside the specific experimental model system employed here.

Our global assessment of protein de novo synthesis upon infection revealed a global reduction in overall protein output during infection with both strains, probably reflecting a global stress response. Also, we observed the well-known shutoff of host protein synthesis[38,40,41,57]. Specific classes such as interferon-related, ribosomal and mitochondrial proteins escaped the shut-off. We observed that the amount of protein synthesis upon infection primarily depends on mRNA levels. Thus, altered translation does not play a major role for the host shutoff, consistent with recent findings[41]. Surprisingly, we also found that viral transcripts were not more efficiently translated than host transcripts. This contrasts with early studies based on reporter systems[57], but corroborates recent ribosome profiling data[41]. Our finding is also consistent with the fact that the codon usage of IAV genes is not optimized to reflect the codon usage of the host[43]. It is also interesting that the translation efficiency of the bird- and human-adapted strains was similarly poor. Thus, adaptation towards high translational efficiency does not seem to be required for crossing the species barrier.

Our unbiased proteomic analysis indicates that the differences between permissive and non-permissive infection depend on differences in viral rather than host protein synthesis. Hence, the orchestrated synthesis of the viral proteome appears to be critically important for permissive infection. This supports the emerging view that modulation of viral protein synthesis underpins host adaptation[58]. Specifically, we find that the strain-specific differences in M1 protein synthesis critically depend on a conserved *cis*-regulatory element, which controls M-segment mRNA splicing. M1 is particularly important for the nuclear export of the viral genome to the cytoplasm[10,11]. Consistently, we observed that the genome of the bird-adapted strain was inefficiently exported (Fig. 3c). We were not able to directly assess the role of M1 for nuclear export because the level of ectopically expressed M1 is much lower than the level reached during infection. Interestingly, we also observed differences in HA and NA expression between both strains (Fig. 3a), which may be relevant in this context.

We found that exchanging only eight nucleotides of the human-adapted M segment to the bird-adapted sequences markedly impaired viral replication. Hence, the *cis*-regulatory element described here plays an important role for host adaptation. However, it is critical to also emphasize that this is not the only relevant factor for IAV host range. For example, despite the overall similar host response, we and others have previously described host factors affecting human and avian virus infections[15–18,20,27]. It is also well-established that the RdRP of avian-adapted strains is less active in human cells[4,44]. Moreover, differences in the binding specificity of viral HAs are known to play an important role for host adaptation[5]. Lastly, M-segment splicing does not only depend on *cis*-regulatory elements but also on *trans*-acting factors, such as NS1, RdRP, NS1-BP, or HNRNPK[49,59–62]. Indeed, while M1 production was clearly impaired in our mutant strain (Fig. 6b), the wild-type bird-adapted strain produced even less (Fig. 3a). Also, the increased production of avian M1 in chicken cells relative to human cells indicates that host factors play a role (Fig. 4f). It is therefore important to interpret our findings in the broader context of viral and host factors that jointly determine the success of IAV replication.

We are living in a pandemic era of IAV infections that began at around 1918[63]. At this time, a virus of ultimately avian origin acquired the ability to spread among humans and later on contributed its genetic material to other pandemic viruses until present. The M gene of this p1918 virus is in some regions similar to bird-adapted sequences, but shows important signatures of mammalian adaptation, especially at the 3′ splice site region[64]. Our results suggest that mammalian adaptation at the 3′ splice site was linked to modulating M segment RNA splicing, which may have been relevant for the emergence of the 1918 pandemics in humans.

## Methods

**Cells and viruses.** A549 (ATCC CCL-185) and HEK293T (ATCC CRL-11268) were grown in Dulbecco's modified Eagle's medium (DMEM) supplemented with 10% (v/v) fetal bovine serum, 2 mM L-glutamine, and antibiotics. Madin–Darby canine kidney (MDCK) type II cells (ATCC CRL-2936) were grown in minimal essential medium supplemented with 2 mM L-glutamine and 10 % (v/v) fetal bovine serum. DF-1 (ATCC CRL-12203) were grown in DMEM supplemented with 10% (v/v) fetal bovine serum and 4 mM L-glutamine. All cells were maintained at 37 °C and 5% $CO_2$. Stocks of the avian influenza virus A/Mallard/439/2004 (H3N2) (Mal) (GISAID accession numbers EPI859640-EPI859647) were grown in the allantoic cavities of 10-day-old embryonated chicken eggs for 2 days at 37 °C. A/Panama/2007/1999 (H3N2) (Pan) (NCBI accession numbers: DQ487333–DQ487340), Pan + Mal M reassortant and Pan-Av mutant virus were grown in MDCK type II cells. Virus stocks were titrated on MDCK type II cells by measuring plaque-forming units (PFUs) or fluorescence-forming units (FFUs). For the latter, cells were infected with different dilutions of virus stock for 5 h. Then, cells were harvested by trypsinization, fixed, and permeabilized by incubation in 75% ethanol for at least 12 h at 4 °C and stained with specific antibody against NP antigen (clone AA5H, Bio-Rad/Serotec, 1:1000, #MCA400). An Alexa Fluor 488-conjugated goat anti-mouse immunoglobulin G (IgG) antibody (Invitrogen) was

used as secondary reagent (1:1000). Cells were analyzed on a FACSCanto II flow cytometer (BD Biosciences) using the FACSDiva software package.

**Cloning and mutagenesis**. Gene fragments containing the coding sequence (nucleotides 29–1007) of segment 7 of A/chicken/Rostock/45/1934 (GenBank accession: CY077423), A/Vietnam/1203/2004 (GenBank accession: HM006762.1), A/swine/Netherlands/25/1980 (GenBank accession: Z26862.1), A/BrevigMission/1/1918 (GenBank accession: AY130766, "p1918"), a mutated A/BrevigMission/1/1918 (with the following point mutations: C718T, A725C, A754G, G760A, G766A, A772G, "p1918 w/ Mal 707–779"), and a mutated version of A/chicken/Rostock/45/1934 (with the following point mutations: G712A, A714G, G740A, C745T, G754A, A760G, "A/chicken/Rostock/45/1934 w/ Pan 707–779") fused to attB1 and attB2 sites were ordered as synthetic double-stranded DNA fragments from Integrated DNA Technologies. The coding sequences of Pan and Mal M segments (nucleotides 29–1007) were amplified from complementary DNA (cDNA) and fused to attB1 and attB2 sites by PCR using the following primers:

fw_Pan: GGGGACAAGTTTGTACAAAAAAGCAGGCTCCAGCCTTCTAAC CGAGGTCG

fw_Mal: GGGGACAAGTTTGTACAAAAAAGCAGGCTCCAGTCTTCTAAC CGAGGTCG

rev_Pan: GGGGACCACTTTGTACAAGAAAGCTGGGTTTACTCCAGCTCTATGCTGA CAAAATG

rev_Mal: GGGGACCACTTTGTACAAGAAAGCTGGGTTTACTCCAGCTCTATGTTGA CAAAATG.

Cloning was done using Gateway Technology (Invitrogen) according to the manufacturer's protocol. Entry clones were generated from pDONR221 vector, expression clones from pDEST26-Flag/HA destination vector. Pan/Mal chimeric constructs were generated by replacing appropriate restriction fragments of wild-type pDEST26-Flag/HA-Pan and pDEST26-Flag/HA-Mal, respectively, with the corresponding synthetic chimeric DNA inserts purchased as gBlocks from Integrated DNA Technologies.

A cDNA copy of the M segment of A/Mallard/439/2004 was amplified by RT-PCR using the following primers:

fw: CGAAGTTGGGGGGGGAGCAAAAGCAGGTAG

rev: GGCCGCCGGGTTATTAGTAGAAACAAGGTAG.

This was followed by insertion into pHW2000 resulting in pHW2000-Mal M. The following point mutations were introduced into pHW2000-Pan M plasmid: A712G, G714A, A740G, T745C, G754A, G760A, G766A, and A772G, creating pHW2000-Pan M-AV. The pHW2000-Mal M plasmid was mutated at the same positions to the respective sequence of the Pan strain (G712A, A714G, G740A, C745T, G754A, A760G, A766G, G772A, "pHW200-Mal M w/ Pan 707–779"). Mutations were introduced using the Gibson Assembly Cloning Kit (NEB) with pDEST26-Flag/HA-Pan w/ Mal 707–779 and the wild-type pHW2000-Pan M or pDEST26-Flag/HA-Mal w/ Pan 707–779 and the wild-type pHW2000-Mal M as templates for PCR. The following primers were used to amplify the insert:

fw_pDest: GAGGACTGCAGCGTAGACGCTTTG

rev_Malw/Pan_pDest: CAATGATACTTGCGGCAATAACGAGAGG

rev_Panw/Mal_pDest: CAAGATCCCAATGATACTCGCAGCAAC.

The following primers were used to amplify the backbone:

fw_Pan_pHW2000: GTTGCTGCGAGTATCATTGGGATCTTG

fw_Mal_pHW2000: CCTCTCGTTATTGCCGCAAGTATCATTG

rev_Mal/Pan_pHW2000: CAAAGCGTCTACGCTGCAGTCCTC.

All constructs were confirmed by cycle sequencing.

**Reverse genetics**. Recombinant IAVs derived from the A/Panama/2007/99 backbone were generated using an eight plasmid system for this strain based on pHW2000 by transfection of human HEK293T cells, followed by passage on MDCK type II cells. For the rescue of the Pan-Mal M reassortant virus, we used pHW2000-Mal M, whereas the Pan-AV virus was generated with pHW2000-Pan-AV together with seven plasmids encoding the other segments of human A/Panama/2007/99 virus.

**In vitro transcription**. First, the M segment region of interest (707–825) that was predicted to be structured was subcloned into pMCh1807_pBS_cdc42E7-boxB (Addgene: 118612) using the primers

Mal_fw: CGATCTCGAGGTCTGAAAGATGATCTTCTTG

Pan_fw: CGATGCTCGAGTCTAAGAGATGATCTTCTTG

Mal/Pan Rev: GCTACGATCGAGTGCAAGATCCCAATG,

and fused to a T3 promoter sequence before being used as template for in vitro transcription with the following primers:

fw: CCAAGCTCGAAATTAACCCTCACTAAAGG

rev: GCTAGCTAGCAGTGCAAGATCCCAATG.

In vitro transcription was carried out using the MegaScript Kit (Life Technologies) according to the manufacturer's instructions with 100 ng of template DNA and subsequent DNA removal. The RNA was purified using phenol/chloroform and resuspended in buffer containing 10 M Tris, 100 mM KCl, 10 mM MgCl$_2$, and incubated at 37 °C for 30 min. To determine differences in RNA

secondary structure, RNA was run on native 1% agarose TAE (Tris-Acetate-EDTA) gels, and then supplemented with ethidiumbromide. To denature secondary structures, the RNA was mixed 1:5 with 65% formamide, 22% formalin, 13% 10× MOPS (3-(N-morpholino)propanesulfonic acid) buffer and incubated at 55 °C for 15 min. Then, samples were run on a denaturing MOPS agarose gel containing 3% formalin and ethidium bromide.

**pAHA-SILAC**. A549 cells were fully labeled in SILAC DMEM (PAA) supplemented with glutamine, 10% fetal bovine serum (Life Technologies) and 2 mM L-glutamine, antibiotics, and with either heavy (R10K8, "SILAC-H"), medium (R6K4, "SILAC-M"), or light (R0K0, "SILAC-L") arginine and lysine (Cambridge Isotope Laboratories). Cells were cultured in SILAC-L/M/H medium for at least six passages. Ten-cm dishes of confluent light-labeled cells were mock-infected, while heavy- and medium-labeled cells were infected with either Pan or Mal strain at a multiplicity of infection (MOI) of 3 PFU (pore-forming unit). Virus was allowed to attach to the cells for 45 min on ice. Cells were washed with pre-warmed phosphate-buffered saline (PBS) before the infection medium was added (SILAC DMEM containing the respective SILAC AA, 0.2% bovine serum albumin (BSA), 2 mM glutamine, antibiotics). Prior to pulse labeling, cells were washed with pre-warmed PBS. Methionine-depleted infection medium additionally containing 100 μM L-AHA (4-azido-L-homoalanine; Anaspec) was added for different 4-h intervals to the cells. Cells were washed in PBS, scraped from the dish, and frozen until further sample processing. Lysis and enrichment for newly synthesized proteins was done using Click-It Protein Enrichment Kit (Invitrogen), with modifications: for the lysis, 283 μl of urea lysis buffer was used per label, samples were incubated for 20 min on ice, sonicated and cell debris was removed before SILAC label were mixed. Ten percent of sample was directly subjected to Wessel–Flügge precipitation and served as the input, and 90% were used for enrichment of newly synthesized proteins. Two hundred microliters of alkyne agarose resin was mixed with 800 μl combined cell lysate and then catalyst solution was added. The enrichment reactions were rotated head over tails for 18 h at room temperature. Enriched proteins were reduced and alkylated as indicated in the manufacturer's instructions. Beads were then washed sequentially (each 5×) in sodium dodecyl sulfate (SDS) wash buffer (supplied with the kit), 8 M urea in 0.1 M Tris/HCl (pH 8.0), 80% acetonitrile in 0.1 M Tris/HCl pH 8.0, and 5% acetonitrile in 50 mM ammonium bicarbonate. Proteins were then digested in 5% acetonitrile/50 mM ammonium bicarbonate overnight using trypsin (Promega). Peptides were then acidified, desalted, and either directly measured on a nano LC-MS/MS setup (see below) or subjected to isoelectric focusing using an OFFGEL fractionator.

Input samples were reduced by adding dithiothreitol (DTT) to a final concentration of 0.1 M and incubation for 5 min at 95 °C. Sulfhydryl groups were alkylated by adding iodoacetamide to a final concentration of 0.25 M and incubation for 20 min in the dark at room temperature. Proteins were precipitated according to Wessel and Fluegge, resuspended in 6 M urea/2 M thiourea and digested into peptides with C-terminal lysine or arginine using Lys-C (3 h) and trypsin (overnight, diluted 4× with 50 mM ABC). Enzyme activity was quenched by acidification of the samples with trifluoroacetic acid. The peptides were desalted with C18 Stage Tips prior to nanoLC-MS/MS analysis.

**pSILAC**. Cells were adapted to SILAC-light medium one day before the experiment and infected as described above using the indicated MOI. Prior to the pulse period, cells were maintained in PBS supplemented with Ca$^{2+}$/Mg$^{2+}$ and 0.2% BSA for 30 min. Then, cells were pulse labeled with SILAC-M or SILAC-H medium for 6 h intervals, harvested, and then combined. Lysis was carried out in 125 mM NaCl, 0.1% SDS, 1% NP-40, 5% glycerol, 50 mM Tris-HCl, pH 7.4 for 1 h on a rotating wheel with subsequent centrifugation. The supernatant was precipitated according to Wessel and Flügge and precipitated proteins were subjected to in-solution digest. Proteins were denatured in 6 M urea/2 M thiourea, reduced, alkylated, and digested using Lys-C (3 h at 20 °C). Then, the digest solution was diluted 4× with 50 mM ammonium bicarbonate buffer and incubated with trypsin (Promega) for 16 h at 20 °C. Afterwards, samples were acidified and subjected to stage tip purification.

**Mass spectrometry**. Peptides from input and AHA-enriched samples were separated on a monolithic silica capillary column (MonoCap C18 High-Resolution 2000, GL Sciences), 0.1 mm internal diameter × 2000 mm length, at a flow rate of 300 nl/min with a 5 to 45% acetonitrile gradient on an EASY-nLC II system (Thermo Fisher Scientific) on an EASY-nLC HPLC (Thermo Fisher) system by 2 or 4 h gradients with a 250 nl/min flow rate on a 15 cm column with an inner diameter of 75 μm packed in-house with ReproSil-Pur C18-AQ material (Dr. Maisch, GmbH). Peptides were ionized using an ESI source on a Q-Exactive, Q-Exactive Plus or a LTQ (linear trap quadrupole) Orbitrap Velos MS (all Thermo Fisher) in data-dependent mode. Q-Exactive and Q-Exactive Plus mass spectrometers were operated in the data-dependent mode with a full scan in the Orbitrap, followed by top 10 MS/MS scans using higher-energy collision dissociation. The full scans were performed with in a m/z range of 300–1700, a resolution of 70,000, a target value of $3 \times 10^6$ ions and a maximum injection time of 20 ms. The MS/MS scans were performed with a 17,500 resolution, a $1 \times 10^5$ target value, and a 60 ms maximum injection time. The LTQ Orbitrap Velos instrument was operated in data-dependent CID top 20 mode. Full scans were performed in

$m/z$ range 300–1700 with a resolution of 60,000 and a target value of $10^6$. MS/MS scans were performed with an isolation window of $2 m/z$ and a target value of 3000.

Peptides from pSILAC samples were separated by 4 h gradients and ionized with ESI source and analyzed on Q-Exactive HF-X instrument (Thermo Fisher) in data-dependent mode. The full scans were performed with a resolution of 60,000, a target value of $3 \times 10^6$ ions and a maximum injection time of 10 ms. The MS/MS scans were performed with a 15,000 resolution, a $1 \times 10^5$ target value and a 22 ms maximum injection time.

**Data analysis**. Raw files of the pAHA-SILAC were analyzed with MaxQuant software version 1.6.0.1 Default settings were kept, except that "requantify" option was turned on. Label-free quantification via iBAQ calculation was enabled. Lys4/Arg6 and Lys8/Arg10 were set as labels and oxidation of methionines, n-terminal acetylation, and deamidation of asparagine and glutamine residues were defined as variable modifications. The in silico digests of the human Uniprot database (downloaded January 2018), the protein sequences of 12 Pan and Mal Influenza virus proteins and a database containing common contaminants were done with Trypsin/P. The false discovery rate was set to 1% at both the peptide and protein level and was assessed by parallel searching a database containing the reverted sequences from the Uniprot database. The resulting text files were filtered to exclude reverse database hits, potential contaminants and proteins only identified by site (i.e., protein identifications that are only explained by a modified peptide). Plotting and statistics were done using R and figures were compiled in Illustrator (Adobe). Raw files for pSILAC were analyzed as described above, except that MaxQuant software version 1.5.2.8 was used. For the pSILAC experiment comparing Mal infection in human and avian cells, spectra were searched against a database additionally containing the uniprot entries of gallus gallus. For the pSILAC experiment comparing Pan and Pan-AV, requantify option was set to off.

**Proteomic data processing**. Two MaxQuant output files were used: proteinGroups.txt and evidence.txt. iBAQ values from infected samples were extracted from proteinGroups.txt. iBAQ values were first normalized by scaling to the iBAQ protein median across all mock-infected samples. This assumes that there are no differences in overall protein synthesis for different mock-infected samples. The iBAQ values were averaged for the corresponding label-swap replicates and proteins were categorized as host or viral. For estimating the newly synthesized protein mass, intensity values of H and M SILAC channels were divided by the summed up intensities of the light channel (mock infected). Data were then averaged for label-swap replicates and summed up for viral and host proteins independently. Finally, data were normalized to the 0–4 h time period.

SILAC ratios of host proteins were processed by first transforming them into log 2 space. The median SILAC-H/L and SILAC-M/L ratios from the input samples were used to estimate the mixing ratio of the input and the H/L and M/L ratios after the enrichment were adjusted correspondingly. SILAC-H/M ratio that relate to the Pan/Mal (or Mal/Pan) infection treatment were normalized to 0. Then, the replicate measurements were averaged. Proteins that were quantified in only one replicate were excluded. Later the top 2% of proteins (highest log2 fold-change) of either Pan/Mock or Mal/Mock condition were selected and multi-set GO enrichment was performed using Metascape tool (http://metascape.org).

Protein level data was matched to RNA level data based on the HGNC official gene symbol. Protein synthesis efficiencies were calculated by subtracting log 10 (RPKM) from log 10(iBAQ) values.

For quantification of pAHA-SILAC viral protein expression kinetics, we extracted all quantifications of peptide level evidences for each individual replicate from evidence.txt. Median log 2 ratios were then normalized to 0 for individual replicates. Comparative viral protein expression kinetics were based on Pan/Mal shared peptides. For each time period, replicate, and viral protein, the Pan/Mal SILAC protein ratio was calculated as the median of all SILAC peptide level ratios. Replicate SILAC protein ratios were averaged. For pSILAC, ratios for viral proteins were extracted from proteinGroups.txt. Non-normalized and log-2-transformed ratios were used.

**RNA-sequencing and data processing**. Total RNAs from A549 cells infected with Pan or Mal strain (infection conditions as described in the pAHA-SILAC experiment) were extracted using TRIzol reagent (Life Technologies) following the manufacturer's protocol. Truseq Stranded mRNA sequencing libraries were prepared with 500 ng total RNA according to the manufacturer's protocol (Illumina). The libraries were sequenced on HiSeq 2000 platform (Illumina) and yielded in total 186 million 101-nt single-end reads. The sequencing reads were first subjected to adapter removal using flexbar with the following parameters: -x 6 -y 5 -u 2 -m 28 -ae RIGHT -at 2 -ao 1 -n 4 -j -z GZ[65]. Reads mapped to the reference sequences of rRNA, tRNA, snRNA, snoRNA and miscRNAs (available from Ensembl and RepeatMasker annotation) using Bowtie2 (version 2.1.0) with default parameters (in --end-to-end & --sensitive mode) were excluded. The remaining reads were then mapped to the human and Pan/Mal influenza A reference genome using Tophat2 (v2.0.10) with parameters: -N 3 --read-gap-length 2 --read-edit-dist 3 --min-anchor 6 --library-type fr-firststrand --segment-mismatches 2 --segment-length 26, and the guidance of RefSeq/Ensembl human gene structure and known viral gene annotation. Gene expression levels (RPKM) were estimated by Cufflinks

(v2.2.1) with parameters: -u --library-type fr-firststrand --overhang-tolerance 6 --max-bundle-frags 500000000. Splice junction reads for various M transcripts were counted with customized Perl scripts. Splice isoforms were accepted that had >500 read counts in both replicates. The same RNA-seq experimental and computational procedures were applied to samples that were transfected, as well as the Pan-AV and the control wild-type infection samples (Pan), except that HiSeq 4000 platform (Illumina) was used and that libraries were sequenced in the $2 \times 151$-nt paired-end mode.

To account for differences in transfection efficiency and sequencing depth from transfected samples, we calculated a scaling factor for NP, PA, PB1, and PB2 across all experiments based on their FPKM expression values. The splice junction read count was then divided by the average of the scaling factors.

**Transfections**. A549 cells were seeded on 6-well plates and transfected with 2.5 μg of expression constructs and Lipofectamine 3000 (Thermo Fisher) reagent according to the manufacturer's instructions. Cells were harvested by trypsinization and subjected to lysis and immunoblotting. For the RdRP reporter experiment, 10 cm dishes of HEK293T cells were transfected with vectors encoding NP (4.5 μg), PA (1.5 μg), PB1 (1.5 μg), or PB2 (1.5 μg) of strain WSN and either pHW2000-Mal M, pHW2000-Pan M pHW2000-Pan-AV, or pHW200-Mal M w/ Pan 707–779 (0.75 μg) using polyethyleneimine. Half of the sample was subjected to RNA isolation and RNA-sequencing 24 h post transfection.

**Lysis, SDS-polyacrylamide gel electrophoresis, and immunoblotting**. Cells were lysed in lysis buffer (125 mM NaCl, 0.1% SDS, 1% NP-40, 5% glycerol, 50 mM Tris-HCl, pH 7.4) for 1 h on a rotating wheel and centrifuged. Supernatant was supplemented with NuPage LDS Sample buffer (Invitrogen), 50 mM DTT, and heated for 10 min at 70 °C. Samples were run on 4–12% Bis-Tris gradient gels (NuPAGE, Invitrogen) before being blotted onto PVDF membrane (Immobilon-P, Millipore) using a wet blotting system (Invitrogen). Specific antibodies against the HA epitope (clone 3F10, Roche, 1:500, #11867423001), vezatin (clone B-1, Santa Cruz, 1:1000, #sc271347), M1 (clone GA2B, Bio-Rad, 1:1000, #MCA401) or M2 (polyclonal, RRID: AB_2549706, Thermo Fisher, 1:1000, #PA5-32233) and suitable horseradish peroxidase-linked secondary antibodies were used. See source data file for raw blot images.

**Immunofluorescence microscopy**. A549 cells were grown on glass coverslips and infected with the indicated viruses at an MOI of 1 FFU (fluorescence-forming unit)/cell. At the indicated time points post infection, cells were fixed for 15 min in 2.5% formaldehyde, permeabilized in 0.2% Triton X-100, and stained with specific antibody against NP antigen (clone AA5H, Bio-Rad/Serotec, 1:1000, #MCA400) in 3% BSA for 1 h. After washing, cells were stained with Alexa Fluor 488-conjugated goat anti-mouse IgG secondary antibody (1:1000) in 3% BSA. Nuclei were counterstained with 4′,6-diamidino-2-phenylindole (DAPI) by adding to the secondary antibody solution. Coverslips were mounted using Mowiol and images were acquired by an Eclipse A1 laser-scanning microscope using the NIS-Elements software package (Nikon). At least 140 cells were counted per condition to quantify the subcellular distribution of NP using the ImageJ software.

**Computational RNA structure analyses**. M segment sequences were obtained from the NIAID Influenza Research Database (IRD) through the website at http://www.fludb.org. We used the following settings for human-adapted strains: date range ≥2009, subtype H3N2, only complete genomes, host human; exclude laboratory strains and duplicate sequences and geographic grouping: South America, Europe, and Asia; for avian-adapted strains: date range ≥2009, only complete genomes, host: avian; exclude duplicate sequences and geographic grouping: Europe and Asia. These two sets of sequences were merged with the respective references sequence, that is, A/Panama/2007/1999—Pan (human) and A/Mallard/439/2004—Mal (mallard), and aligned using the program Muscle[66] resulting in two MSAs, one for the human-adapted strains comprising 403 sequences and one MSA for the avian-adapted strains comprising 199 sequences. Both alignments are straightforward to establish based on the long open reading frame of the M1 isoform covering almost all of the M segment and on the overall high primary sequence conservation of the M segment. Evolutionary trees relating the sequences in either MSA were then derived using PhyML in conjunction with the HKY evolutionary model, which provided the best fit to the data [version v3.0[67]].

These two input alignments (including the combined annotation of the known protein-coding M1 and M2 regions) and the corresponding evolutionary trees were then used as input to RNA-Decoder[52]. We used RNA-Decoder to identify the RNA secondary structure that is best supported by the evolutionary signals contained in the two input alignments (the so-called maximum-likelihood structure). The predictions by RNA-Decoder also included the posterior base-pairing probabilities for each base pair of the predicted RNA structure. Predicted base pairs with a base-pairing probability smaller than 25% were omitted from the RNA structure visualization. Note that each multiple sequence alignment was analyzed by RNA-Decoder in one chunk, that is, without partitioning it artificially into sub-alignments.

Finally, the predicted RNA structure element nucleotides 733–766 was plotted with the sequence of the Mal strain using VARNA tool[68] and the RNA structures predicted for the two alignments of avian- and human-adapted sequences was visualized using R-chie[69], including information on the pairing probability of each base pair.

**Nucleotide-level conservation**. To retrieve M segment consensus sequences for human and avian-adapted strains, we used the NIAID IRD[70] through the website at http://www.fludb.org (downloaded 10 July 2018). The following options were applied to retrieve avian-adapted consensus sequences: date range: 1999–2011, complete genome, minimum length M segment sequence 1027, and exclude duplicated and laboratory strains. The following options were used to retrieve the human consensus: date range: 1999–2011, complete genome, minimum length M segment sequence 1002, and exclude duplicated and laboratory strains.

**Reporting summary**. Further information on research design is available in the Nature Research Reporting Summary linked to this article.

## Data availability

Proteomic data relating to the pAHA-SILAC and pSILAC experiments was uploaded to ProteomeXchange Consortium via the PRIDE partner repository with the dataset identifier PXD011321 (pAHA-SILAC) and PXD015475, PXD015474 (pSILAC) RNA-sequencing data is publicly available under PRJNA495615 [https://www.ncbi.nlm.nih.gov/sra/PRJNA495615]. The source data underlying Figs. 1c, d, 2a–d, f–h, 3a, d, 4a, b, d, f, 5c–g, 6b–f and Supplementary Figs. 2 and 4a–c are available as Source Data file.

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

## Acknowledgements

This work was supported by the German Ministry of Education and Research (Virosign grant 0316180B). We would like to acknowledge the excellent technical assistance of Gudrun Heins. We thank Katrina Meyer for providing the pDEST26-Flag/HA plasmid and Koshi Imami for setting up the nLC system using monolithic columns. We also thank Marina Chekulaeva for in vitro transcription constructs and Sascha Sauer of the MDC Scientific Genomics Platform for RNA-sequencing.

## Author contributions

B.B. performed most of the experiments and data analyses. K.E. established the AHA pulse labeling technology and generated pAHA-SILAC data together with B.B. A.S. and K.P. performed the infection experiments for proteomic analysis supervised by T.W. X.W. and J.H. generated and analyzed RNA-seq data supervised by W.C. I.H. performed qRT-PCR analysis supervised by L.W. and B.B. M.H., L.W., and B.V. contributed to cloning and transfection of reporter constructs. I.M.M. performed the RNA structure analyses. M.B. conducted reverse genetic experiments. T.W. and M.S. conceived and supervised the study. B.B., L.W., T.W., and M.S. interpreted the data. B.B. and M.S. wrote the manuscript with input from all authors.

## Competing interests

The authors declare no competing interests.
