## [Peer Review File · Nature Communications]

Reviewers' Comments:

Reviewer #1:

Remarks to the Author:

This manuscript seeks to understand the molecular determinants of host restriction of influenza A virus (IAV). This question is significant as it is part of understanding the difference between pandemic vs normal infections of IAV. The authors here test the hypothesis that differential expression of proteins in infected cells may lay at the core of the different replication rates. Specifically, they compare the proteome of cells infected with bird-adapted vs. human-adapted strains of IAV. From this data they conclude that a major difference between these two strains is expression of the viral M1 protein.

In general this study is well executed and provides datasets that are well controlled and analyzed, and thus will likely be useful to other researchers. In addition, the conclusion that M1 expression is differentially regulated in distinct strains of IAV is of interest. However, a few issues need to be addressed in order to fully support the conclusions of the manuscript

1. In addition to differences in M1, the proteomic analysis also reveals a difference in expression of HA. Does this difference also contribute to functional differences between the IAV strains?
2. The authors propose that the reduction of M1 leads to reduction of NP nuclear export, however this is not experimentally tested. Does ectopically induced expression of M1 in MAL-infected cells reverse the NP export defect?
3. The conclusion that a structure in the M1 3' splice site controls efficiency of splicing needs to be better supported. (1) the sequence of both the human and avian consensus sequences around the 3' splice site need to be shown – not just a few nucleotides around the avian sequence. (2) while the GG is in a single stranded region in the proposed structure – the structure would hinder binding of the spliceosomal proteins (U2AF1, U2AF2, the U2 snRNP) and thus would be predicted to repress, not enhance, splicing. (3) to say that a structure is important one has to make mutations to disrupt the structure and then compensatory mutations to restore it and test function.
4. The authors never shown whether it is the decrease in M1 or the increase in M2 that hinders pathogenesis of the avian strain. This can easily be tested by KD of M2 or ectopic expression of M1 in infected cells with a vector or modified IAV -derived M1 cDNA.
5. In figure 6C, why does M2 differ in size but not abundance? Is actually an incorrect version of M2 made in one of these strains?
6. Authors need to clarify whether or not overall M segment RNA levels (M1 and M2 total) are changed when swapping the Avian 3' splice site into the human M segment (Fig. 6d and e). If the avian-fusion M segment is transcribed differently than the WT human segment, the authors should mention this and reconcile it with their model. The current data does not rule out that the alterations in splicing could be a result of changes in RNA synthesis rates.

Reviewer #2:

Remarks to the Author:

The work by Bogdanow et al. "The dynamic proteome of Influenza A virus infection identifies M segment splicing as a host range determinant" describes the differences at the proteome level between human cells infected with human-adapted and bird – adapted IAV strains and identifies the important differences in splicing of the viral matrix protein M1 between the two strains. I was asked to comment on the proteomic aspects of the manuscript.

The authors used the method they are experts on, the metabolic pulse labeling that identifies newly synthesized proteins, and quantitative shotgun proteomics. This is a very solid and meticulously carried out work. I think it would be of interest not only to the proteomics field, but also to the

infectious disease and virology communities. Therefore, I recommend this paper for publication in Nature Communications. I have a few minor comments that should be easily addressed.

1. It would be helpful if the authors clarified the labeling scheme a little better. For each time point, one of the light/medium/heavy labeled cell batches was infected either with Pan, Mal, or mock infected. This was varied to account for labeling bias but it may be not entirely clear for the people who are reading the paper and looking at the figure.
2. Also, Figure 1 has colors that are not very helpful: in (a), and (b), green and blue (dots and text) are hard to distinguish. In (c) everything is blue. It may be worth the time to change the color scheme.

Aleksandra Nita-Lazar

Reviewer #3:

Remarks to the Author:

This study by Bogdanow et al. extends previous work from the same collaborative consortium, which aimed to identify differentially regulated cellular factors in the human alveolar A549 cell line infected with a human seasonal H3N2 influenza virus (Pan) versus an avian low-pathogenic H3N2 influenza virus (Mal), using Spike-in SILAC mass spectrometry-based quantitative proteomics (Sadewasser et al, Mol Cell Proteomics 2017). This initial study provided steady-state measurements. Here the authors used pulse SILAC and quantitative mass spectrometry to compare the human proteome dynamics in A549 cells infected with the human Pan or the avian Mal influenza virus. They report similar synthesis profiles of host cell proteins in both conditions, suggesting that viral-induced modulation of translation is not a major determinant of host-range. In contrast, their data point to a lower M1 to M2 ratio in cells infected with the avian virus, which they attribute, at least partially, to an increased splicing of the M1 mRNA.

This study contributes to advance our understanding of influenza virus-induced shut-off of cellular proteins expression. However, it does not strongly support the author's claim that M segment splicing pattern is a host-range determinant. The model proposed in Figure 7 remains largely speculative and has no mechanistic justification. There are a number of unclear and unconvincing parts in the manuscript, which are summarized below.

(1) Page 7, the authors state that SILAC-based relative quantification of de novo protein synthesis shows "overall good reproducibility". As only two biological replicates were performed, statistical significance could not be assessed. Therefore it would be more appropriate to mention the Spearman's correlation coefficients of the two biological replicates (0.81, 0.51, 0.86 and 0.71 for the 0-4h, 4-8h, 8-12h and 12-16h samples respectively, as shown in Supplementary Figure 1). The use of Spearman's rather than Pearson's correlation coefficient should be justified. Scatter plots of the biological replicates would be the best way to document the reproducibility of the data.

(2) Following up with the previous comment, it should be systematically stated in the figure legends whether the data are representative of one replicate or both (the information is missing in several instances, e.g. in Figure 2, 3 and 4). Likewise, the number of biological replicates should be indicated in the legends of Figures 5c, 5d, 6b-e.

(3) In Figure 2e, the scatter plot should be shown for both the Mal.mock and Pan.mock datasets. In Figure 2f, the exact nature of the IBAQ values (which time period) and RPKM value (which time point) used to calculate the IBAQ / RPKM ratios should be explained. If IBAQ 8-12h and RPKM 8hpi were used, as suggested from the data shown in Figure 2d and 2e, this would correspond to a late time

point in the course of infection and may not allow to conclude that viral proteins are, overall, “less efficiently synthesized than host proteins” (page 9).

(4) In the discussion, the findings about the human proteome dynamics in infected A549 cells should be more thoroughly compared to and integrated with previously published proteomics studies on influenza-virus infected cells, including the authors initial comparative study using the same human Pan and avian Mal viruses as in the present study. The added value of the pulse SILAC compared to the Spike-in SILAC dataset should be better highlighted.

(5) Figure 3. The SILAC protein profile of the NS1 protein could not be determined in Figure 3a, because of a lack of shared peptides between the Pan and the Mal samples. However the authors should point out the fact that in their previous study, Mal-infected cells showed not only decreased steady-state levels of M1 protein but also increased levels of NS1 protein, compared to Pan-infected cells (Figure 1B in Sadewasser et al, Mol Cell Proteomics 2017). Therefore the production of NS1/NS2 mRNAs and proteins might well be imbalanced in Mal-infected cells, and this point should be further documented. This point is particularly relevant to the delayed nuclear export of vRNPs illustrated in Figure 3c-3d, as NS2 plays an essential role in the nuclear export of viral vRNPs. In Figure 3, it would be informative to stain the cells for M1 and NS2. Complementation experiments would be needed to demonstrate that the amount of M1 protein (or possibly, NS2 protein) is limiting.

(6) The M2/M1 and NS2/NS1 mRNA ratios that can be estimated from the histograms shown in Figure 4a are surprisingly different from those published previously by several teams. In particular the NS2/NS1 ratio is much higher than the one reported upon RNAseq analysis of H3N2 influenza virus-infected cells (in the 0.5-5% range, Fabbozi et al, Journal of Virology, 2018) or upon RT-qPCR analysis of the A/WSN/33 laboratory strain (in the 10-15% range, Robb et al J Gen Virol 2009, Fournier et al PLoS Path 2014).

If RPKM for NS2 were measured at the splice junction whereas RPKM for NS1 were measured on the whole length of the mRNA, this might introduce a bias in the evaluation of the NS2/NS1 ratio because of irregular coverage. The authors should provide additional details about how RNAseq data were processed, indicate the M2/M1 and NS2/NS1 ratios as determined by RT-qPCR analysis (which is uneasy to infer from the type of graphs shown in Figure 4e), and discuss their findings with regard to the literature.

(7) The data shown in Figure 5 and Figure 6 indicate that the sequence next to the 3' splice site of the M segment can determine a higher splicing efficiency in Mal-infected than in Pan-infected cells. However they do not convincingly demonstrate that this sequence is involved in mammalian adaptation. To this end, it would have been more relevant to generate a mutant Mal virus in which the 3' splice site region of M is mutated towards the corresponding Pan sequence. In addition, the M splicing pattern upon infection with the Mal virus should have been compared in human and in avian cells.

(8) It is unclear why the authors did not take into account a larger 3' splice site region than the one highlighted in figure 5e, to include the adjacent secondary structure which is predicted to occur in the human but not in the avian virus (all the more so as the latter is predicted with a higher probability than the secondary structure on which the authors chose to focus).

(9) To better assess the global significance of the strain-specific splicing patterns with respect to host-range, the analysis should be extended to additional human and avian viral isolates. In particular, the Mal virus used in this study might not be representative of all avian viruses, as it does not show a marked and global defect in the production of viral mRNAs and proteins at early time-points upon infection of human cultured cells, as reported in the literature for several avian influenza viruses.

Other comments.

(10) In the introduction, the information about expression of viral proteins from the M segment should be improved. The function of the M1 and M2 proteins is approximatively described. Important references about the ratio of spliced to unspliced mRNAs are missing (e.g. Robb & Fodor J Gen Virol 2012 , Tsai et al PLoS Pathogens 2013). More generally, a number of references about influenza virus biology in the introduction could be updated (e.g. Ref #7 Fodor 2013 could be replaced with the more recent te Velthuis and Fodor 2016).

(11) The authors should clarify whether the M segment sequence alignments were performed on H3N2 isolates only, as stated in the Results section page 14, or on H3N2 and pH1N1 isolates, as stated in the Methods section page 37. In both cases, the rationale should be explained.

(12) Figure 6. Panel a, it should be indicated whether the nucleotide changes result in amino acid changes.

(13) Analysis of M1 and M2 mRNA levels by qRT-PCR. Was the M1/M2 specificity assessed ?

(14) Page 21, the statement « we are living in a pandemic era of IAV infections that began at around 1918 » is misleading, as it may suggest that there have been no influenza pandemics before 1918.

Point by point response

We thank all three reviewers for carefully reading our manuscript and for their valuable and constructive comments! We are pleased to read that they feel our study is “well executed”, “well controlled and analyzed” (reviewer #1), “very solid and meticulously carried out” (Aleksandra Nita-Lazar) and “contributes to advance our understanding of influenza virus-induced shut-off” (reviewer # 3)

We carefully addressed the comments made by the reviewers and performed a number of additional experiments. While this took us longer than we had hoped, we are very happy with the results we obtained since they further strengthen the manuscript and corroborate our conclusions:

- 1. We infected an avian cell line with the bird-adapted Mal strain (Figure 4E,F). This experiment showed that the bird-adapted strain produces more M1 in avian cells than in mammalian cells. Hence, the reduced M1 production of avian viruses in human cells appears to reflect poor adaptation to the mammalian splicing environment.*
- 2. We biochemically characterized the RNA secondary structure of the M segment using in vitro transcription and non-denaturing gel electrophoresis (Figure 5G). This experiment showed that (i) the Mal and Pan strain M segments show biochemically distinguishable secondary structures and that (ii) this difference is due to the splice site region.*
- 3. We used a minireplicon assay to assess if the differences in M segment RNA levels might be due to differences in primary transcription rather than due to differences in splicing (Figure 5D). We found that the differences depend on splicing and the splice site region, corroborating our conclusion.*
- 4. We characterized mRNA levels in cells infected with the Pan-Av mutant strain using RNAseq (Figure 6D,E). This experiment confirmed that the point mutations in the splice site region affect M1 protein levels by impacting splicing.*
- 5. We included M segments of additional IAV strains (Supplementary Figure 4C). This further supported our conclusion that M segment splicing is systematically different in avian-adapted versus human-adapted strains, supporting the general relevance of our findings beyond the specific strains analyzed in our paper.*

A detailed point by point response is appended below. We feel that these experiments markedly improved the quality of our manuscript and think that it is now ready for publication in Nature Communications.

Reviewer #1:

This manuscript seeks to understand the molecular determinants of host restriction of influenza A virus (IAV). This question is significant as it is part of understanding the difference between pandemic vs normal infections of IAV. The authors here test the hypothesis that differential expression of proteins in infected cells may lay at the core of the different replication rates. Specifically, they compare the proteome of cells infected with bird-adapted vs. human-adapted strains of IAV. From this data they conclude that a major difference between these two strains is expression of the viral M1 protein. In general this study is well executed and provides datasets that are well controlled and analyzed, and thus will likely be useful to other researchers. In addition, the conclusion that M1 expression is differentially regulated in distinct strains of IAV is of interest. However, a few issues need to be address in order to fully support the conclusions of the manuscript.

We thank the reviewer for the supportive and encouraging comments!

1. In addition to differences in M1, the proteomic analysis also reveals a difference in expression of HA. Does this difference also contribute to functional differences between the IAV strains?

This is indeed a relevant point. Importantly, while our manuscript reports that M segment splicing is a host range determinant, we do not claim that it is the only factor. We stressed this throughout our manuscript, for example:

- *Title: The dynamic proteome of influenza A virus infection identifies M segment splicing as a host range determinant*
- *Abstract: Thus, our data identifies M segment RNA splicing as a viral determinant of host range.*
- *Discussion: "... it is critical to also emphasize that this is not the only relevant factor for IAV host range. For example, despite the overall similar host response, we and others have previously described host factors affecting human and avian virus infections. It is also well-established that the RdRP of avian-adapted strains is less active in human cells. Moreover, differences in the binding specificity of viral hemagglutinins (HA) are known to play an important role for host adaptation. Lastly, M-segment splicing does not only depend on cis-regulatory elements but also on trans-acting factors, such as NS1, RdRP, NS1-BP or HNRNPK. Indeed, while M1 production was clearly impaired in our mutant strain (Figure 6B), the wild-type bird-adapted strain produced even less (Figure 3A). It is therefore*

important to interpret our findings in the broader context of viral and host factors that jointly determine the success of IAV replication.

*Having said this, the observed difference in HA expression are indeed interesting, especially since membrane accumulation of HA in infected cells has been shown to promote MAPK signalling (Marjuki et al. 2006, JBC). We therefore assessed whether MAPK targets are differentially expressed in A549 cells infected with Pan or Mal strains in our data using recently published MAPK-responsive genes in A549 cells (Yue et al., 2017, Genes Dev). Specifically, we assessed how genes that are MAPK transcriptionally responsive and genes that are transcriptionally inert to MAPK (“control genes”, both subsets from Yue et al., Supplemental Table 6) are expressed during Pan or Mal infection (see **Figure R1**). We observed that MAPK responsive genes were stronger expressed under Pan than Mal infection. Hence, this analysis does not support increased HA-induced MAPK activation by the Mal strain relative to Pan.*

Of course, this analysis does not rule out the possibility that differences in HA expression play a role. For example, since the balance between HA and NA is critical due to their complementary activities (see Gaymard et al., 2016, Clin Microbiol Infect for a review), it might be that diverging kinetics of HA/NA levels contribute to the species barrier. To confirm this it would be necessary to measure HA binding affinity and NA enzymatic activity, which we feel is beyond the scope of our manuscript.

Since we concentrated on M1 splicing, we think that adding additional text/figures on the possible role of HA would be distracting. However, to stress even more that additional factors are also relevant, we added the following sentence to the discussion:

“Interestingly, we also observed differences in HA and NA expression between both strains (Fig. 3 A) that may be relevant in this context.”

Figure R1. Relative mRNA levels of MAPK responsive genes (Yue et al., *Genes Dev*, 2017) and control genes comparing Pan and Mal infection (*p*-value Mann-Whitney test: 2.4e-09).

2. The authors propose that the reduction of M1 leads to reduction of NP nuclear export, however this is not experimentally tested. Does ectopically induced expression of M1 in MAL-infected cells reverse the NP export defect?

*We also thank the reviewer for this important question. In our manuscript we took care to emphasize that “non-permissive infection **correlates** with reduced M1 production and impaired nuclear export of NP”. While we hypothesize that low M1 levels may be causative for nuclear retention of NP, we did not experimentally test this. We base this hypothesis on the very well established role of M1 in nuclear export of vRNPs. Specifically, we cited papers by the Helenius lab (references 15 and 16) that show M1 as a crucial mediator of vRNP export.*

The experiment suggested by the reviewer is complicated by the low transfection efficiency of A549 cells. Therefore, in an effort to perform the experiment suggested, created a stable A549 cell which inducibly expresses M1 from the human-adapted strain

(Figure R2 A). Unfortunately, the M1 expression level that can be achieved in this cell line is much lower than the M1 level that is achieved during IAV infection (Figure R2 B). Consequently, the additional ectopic expression of M1 does not substantially contribute to the overall cellular pool of this protein. As expected, the low level of additional ectopic M1 expression did not result in a visible increase of NP export (Figure R2 C). Hence, directly testing the impact of M1 levels on NP export in our system is not feasible. To acknowledge this we added the following sentence to the discussion:

“We were not able to directly assess the role of M1 for nuclear export because the level of ectopically expressed M1 is much lower than the level reached during infection.”

a

b

c

Figure R2. Ectopic M1 expression. (A) We generated a stable A549 cells that inducibly expresses M1 upon addition of doxycycline. (B) Infecting the stable cell line with the Mal strain reveals that the expression level of M1 that is achieved during infection is much higher than the level of ectopically expressed M1, MOI=1 (PFU/cell). (C) Immunofluorescence microscopy shows that the low amount of ectopically expressed M1 has no significant effect on the NP export upon infection with the Mal strain.

3. The conclusion that a structure in the M1 3' splice site controls efficiency of splicing needs to be better supported. (1) the sequence of both the human and avian consensus sequences around the 3' splice site need to be shown – not just a few nucleotides around the avian sequence. (2) while the GG is in a single stranded region in the proposed structure – the structure would hinder binding of the spliceosomal proteins (U2AF1, U2AF2, the U2 snRNP) and thus would be predicted to repress, not enhance, splicing. (3) to say that a structure is important one has to make mutations to disrupt the structure and then compensatory mutations to restore it and test function.

We thank the referee for her/his valid points. Proving that an RNA structure directly controls splicing is extremely difficult: Splicing occurs in a given cellular context and involves numerous factors such as RNA-binding proteins, other RNAs, RNA and protein modifications, transcription rates, cis-regulatory RNA elements etc.. Therefore, we refrained from concluding that the secondary structure controls splicing. Instead, we wrote “that the M segment of avian and human-adapted [...] isolates contain evolutionarily conserved RNA secondary structures that markedly differ in exactly the region that is critical for strain-specific splicing.”

*Nevertheless, we fully agree with the reviewer that the predicted existence of different RNA secondary structures should be validated by orthogonal methods. We therefore cloned the region of the M segment containing evolutionarily conserved secondary structure (nt 707-825) from both strains into a plasmid for in vitro transcription by T3 polymerase (backbone construct: Addgene 118612). Then we PCR amplified the sequence and promoter region and used the amplicon as a template for in vitro transcription. We purified the in vitro transcribed RNA and ran it on native and denaturing agarose gels, which were stained with ethidiumbromide (see **Figure R3**). Indeed, we observed different migration behavior for Pan and Mal under native but not denaturing conditions, consistent with different secondary structures in vitro. Importantly, when substituting 8 nucleotides in the region 707-779 (termed 3' splice site region in our manuscript) from Mal into the Pan backbone (lane 3) we observed the same migration pattern as for the Mal wildtype RNA. Conversely, when substituting 8 nucleotides from the Pan into Mal backbone (lane 4) we observed the same migration*

pattern as for the Pan wildtype RNA. This experiment supports our conclusion that differences in secondary structure exist. Importantly, this still does not prove that these structural differences cause differential splicing, and we were therefore careful not to make this claim in the paper. We added the new results to Figure 5.

Figure R3. Differences in RNA secondary structure at the 3' splice site. The indicated *in vitro* transcribed RNAs were loaded on denaturing or native agarose gels.

To answer the specific comments made by the reviewer:

(1) Our analysis of putative RNA secondary structures in Figure 5 is based on many sequences. It is important to note that we cannot consider all individual sequences that are available since this would be computationally too extensive. For this reason we selected a representative set of sequences as outlined in the Methods section. We now included an alignment of the Pan and Mal sequences and corresponding consensus sequences to the supplement (**Figure R4** below). This alignment shows that avian and human sequences diverge by 5 nucleotides in the region of interest (pos: 745, 754, 760, 766 and 772), all of which are retained with the two strains that we investigated (Pan and Mal). Thus, the Pan and Mal strains are by and large representative for human and

avian strains in this region. We added this alignment as supplementary Figure S4 and corresponding text to the results section.

Figure R4. Alignment of consensus sequences of avian and human 3' splice site regions.

(2) The referee is right, the GG is not base-paired in our prediction. However, we are not certain whether a structured 3' splice site would be expected to enhance or inhibit splicing. In fact, the literature provides not only examples where a structured 3' splice site promotes splicing (for example Coleman and Roesser, 1998, *Biochemistry*), but also where a structured splice site inhibits splicing (Watakabe et al. 1989, *Nucleic Acids Research*).

(3) The new analyses presented above (Fig. R2) confirm that differences in secondary structure do exist, and that they are due to specific differences in the 3' splice site region. Also, infections with a Pan virus carrying the avian splice site region shows that this region is important for permissive infection (Figure 6). Nevertheless, the reviewer is right in pointing out that this does not prove that the structure per se is relevant for host range. We would have loved to introduce “compensatory” mutations into the avian splice site region which would disrupt the avian hairpin and restore the human secondary structure. Unfortunately, this is not possible: The splice site region contains two ORFs with different reading frames (for M1 and M2) which makes it impossible to introduce silent mutations that would alter the secondary structure in the intended manner. Essentially, the bases in this region are “fixed”, and there is no option to play around with them without introducing stop codons or amino acid changes. Again, this is one reason why we refrained from making the claim that the differences in secondary structure are directly responsible for host range.

4. The authors never shown whether it is the decrease in M1 or the increase in M2 that hinders pathogenesis of the avian strain. This can easily be tested by KD of M2 or ectopic expression of M1 in infected cells with a vector or modified IAV -derived M1 cDNA.

This would be a relevant point if M2 levels were indeed so different. However, in stark contrast to M1, we see rather similar M2 mRNA levels upon infection with both strains in

the RNAseq data (Figure 4D, less than 2-fold more M2 for Mal strain, versus 6.5 fold less M1 for the Mal strain). We conclude that the rather similar M2 levels are probably due to two opposing effects. On the one hand, the polymerase of the avian strain is less active in mammalian cells, thus producing lower levels of M segment vRNAs/mRNAs. On the other hand, the increased splicing increases the M2/M1 ratio. In combination, both effects compensate each other and result in rather similar M2 levels. We would like to draw the attention of the reviewer to corresponding text in the results section:

“In contrast, M2 mRNA levels were rather similar. We note that the comparable M2 mRNA level during non-permissive infection results from two opposing processes -- the increased splicing of the primary transcript to the M2 mRNA, and the global reduction in viral transcripts, which is probably due to the impaired polymerase activity^{8,9}.”

5. In figure 6C, why does M2 differ in size but not abundance? Is actually an incorrect version of M2 made in one of these strains?

This is an interesting point. To assess whether an incorrect version of M2 is made we took a detailed look at all peptides we observed.

The following peptide sequences were identified in AHA-dataset for M2:

Pan

AA 79-97 EQQNAVDADDSHFSIELE

AA 2-18 SLLTEVETPIRNEWGCR

AA 2-12 SLLTEVETPIR

Mal

AA 78-97 QEQQSAVDVDDGHFVNIELE

AA 2-18 SLLTEVETPTRNGWECK

AA 2-12 SLLTEVETPTR

These peptide sequences match to the expected coding sequence of the virus.

The following peptide sequences for M2 in the Pan-AV mutant were identified in our pSILAC run:

Pan-AV

AA 2-12 SLLTEVETPTR

AA 2-12 (ac)SLLTEVETPTR

Again, the peptide sequences match the expectations. Thus, we see no evidence of an incorrect version of M2 being made by one of these strains. The differences in the running behavior of M2 between these two strains is probably due to the 5 AA that are different between the two mutant viruses in the M2 ectodomain.

6. Authors need to clarify whether or not overall M segment RNA levels (M1 and M2 total) are changed when swapping the Avian 3' splice site into the human M segment (Fig. 6d and e). If the avian-fusion M segment is transcribed differently than the WT human segment, the authors should mention this and reconcile it with their model. The current data does not rule out that the alterations in splicing could be a result of changes in RNA synthesis rates.

*To directly address if the Pan-AV M segment is transcribed differently, we performed Minigenome replication assays. To this end, we transfected HEK293T cells with expression vectors for PB1, PB2, PA and NP (of WSN strain). In addition, we transfected Pan and Mal M wildtype and chimeric segments (pHW2000). In this system, the accumulation of M segment derived mRNAs depends primarily on the activity of the IAV/WSN polymerase. We then systematically investigated the levels of M derived RNAs using RNAseq (see **Figure R5**). We observed that mRNA levels were not lower for cells transfected with the Pan-AV construct (size of the pies). Thus, the mutant M segment is not transcribed less efficiently. Nevertheless, the differences in abundance of the various isoforms are consistent with the differences we observed during infection. These results indicate that the differences in the abundance of the isoforms between strains are not due to differences in primary transcription but rather due to differential splicing, further corroborating our data. We added these data to Figure 5.*

*We also extended this analysis to infection experiments with the Pan and the Pan-Av strains and carried out new RNAseq analyses (**Figure R6**). In these data, the Pan-Av strain showed increased splicing to alternative isoform, consistent with the proteomic data (Fig. 6b). As expected for an attenuated infection, the total number of viral transcripts (8 hpi) is lower upon Pan-Av infection (size of pie charts in **figure R6 B**). However, as the previous experiment shows, this does not appear to result from differences in primary transcription. We included these new data in Figure 6 E.*

Figure R5. The 3' splice site region controls M segment RNA splicing in a RdRP reporter system. HEK293T cells were transfected with vectors encoding NP, PA, PB1, PB2 (WSN strain) as well as wild-type and chimeric M segments. (Mal-Hu corresponds to the avian-adapted M segment with a human 3' splice site region). Shown are splice junction reads for the various M segment RNA isoforms that were normalized to the levels of the polymerase. The area of the pies scales to the total number of normalized splice junction reads.

Figure R6. The 3' splice site region controls M segment RNA splicing A549 cells were infected with Pan and Pan-AV IAV strains at an MOI of 1. At 8 hours post infection, cells were harvested, RNA was isolated and subjected to RNAseq. **(a)** Log₂ fold-changes in mRNA levels (FPKMs) for the indicated viral transcripts comparing Pan and Pan-AV infection (average of two replicates). **(b)** Shown are splice junction reads for the various M segment RNA isoforms. The area of the pies scales with the total number of splice junction reads.

Reviewer #2:

The work by Bogdanow et al. “The dynamic proteome of Influenza A virus infection identifies M segment splicing as a host range determinant” describes the differences at the proteome level between human cells infected with human-adapted and bird – adapted IAV strains and identifies the important differences in splicing of the viral matrix protein M1 between the two strains. I was asked to comment on the proteomic aspects of the manuscript.

The authors used the method they are experts on, the metabolic pulse labeling that identifies newly synthesized proteins, and quantitative shotgun proteomics. This is a very solid and meticulously carried out work. I think it would be of interest not only to the proteomics field, but also to the infectious disease and virology communities. Therefore, I recommend this paper for publication in Nature Communications. I have a few minor comments that should be easily addressed.

We thank Aleksandra Nita-Lazar for her supportive comments on our methodology and the results we obtained!

1. It would be helpful if the authors clarified the labeling scheme a little better. For each time point, one of the light/medium/heavy labeled cell batches was infected either with Pan, Mal, or mock infected. This was varied to account for labeling bias but it may be not entirely clear for the people who are reading the paper and looking at the figure.

We apologize for not having described this in sufficient detail. Indeed, we performed two completely independent biological replicates of the AHA-SILAC experiments with swapped isotope labels. For data analysis, we combined the data from both label swap experiments as described in the methods section. The cartoon in figure 1 A does not depict this in detail because we feel this would be confusing. We now included the following text to the legend of figure S1 (which shows the correlation between label swap experiments):

“In the first replicate SILAC H labelled cells were infected with the Mal strain, SILAC M labelled cells infected with the Pan strain, and SILAC L cells were mock infected. In the second replicate, SILAC M labelled cells were infected with the Mal strain, SILAC H labelled cells were infected with the Pan strain and SILAC L cells were mock infected.”

2. Also, Figure 1 has colors that are not very helpful: in (a), and (b), green and blue (dots and text) are hard to distinguish. In (c) everything is blue. It may be worth the time to change the color scheme.

This is a very good suggestion. We changed the color scheme in figure 1 to avoid confusion.

Aleksandra Nita-Lazar

Reviewer #3:

This study by Bogdanow et al. extends previous work from the same collaborative consortium, which aimed to identify differentially regulated cellular factors in the human alveolar A549 cell line infected with a human seasonal H3N2 influenza virus (Pan) versus an avian low-pathogenic H3N2 influenza virus (Mal), using Spike-in SILAC mass spectrometry-based quantitative proteomics (Sadewasser et al, Mol Cell Proteomics 2017). This initial study provided steady-state measurements. Here the authors used pulse SILAC and quantitative mass spectrometry to compare the human proteome dynamics in A549 cells infected with the human Pan or the avian Mal influenza virus. They report similar synthesis profiles of host cell proteins in both conditions, suggesting that viral-induced modulation of translation is not a major determinant of host-range. In contrast, their data point to a lower M1 to M2 ratio in cells infected with the avian virus, which they attribute, at least partially, to an increased splicing of the M1 mRNA. This study contributes to advance our understanding of influenza virus-induced shut-off of cellular proteins expression.

We thank the reviewer for carefully reading our manuscript and for his/her constructive comments. We also think that our data contributes to advance our understanding of the IAV-induced host shutoff.

However, it does not strongly support the author's claim that M segment splicing pattern is a host-range determinant. The model proposed in Figure 7 remains largely speculative and has no mechanistic justification. There are a number of unclear and unconvincing parts in the manuscript, which are summarized below.

*We thank the reviewer for carefully reading our manuscript and for her/his valuable feedback. The model depicted in figure 7 is indeed speculative, which is why we refer to it as a "hypothetical model" both in the text and in the figure legend. We would also like to point out that we do not claim that M segment splicing is the **only** or **the most critical factor** for IAV host range. In fact, we explicitly wrote that "it is critical to also emphasize that this is not the only relevant factor for IAV host range", and we highlight a number of previously identified factors. Nevertheless, we are confident that our work does support the claim that M segment splicing is a host range determinant: On the one*

hand our experiments with reporter construct clearly establish the relevance of the splice site region for M segment splicing. On the other hand, our infection experiments with recombinant viruses show that the M segment and specifically the splice site region is a relevant host range factor (Fig. 6f): Pan strains carrying the M segment of the Mal strain are markedly attenuated, and the same effect is also seen when only exchanging 8 nucleotides from the Pan splice site regions into the corresponding Mal residues.

(1) Page 7, the authors state that SILAC-based relative quantification of de novo protein synthesis shows “overall good reproducibility”. As only two biological replicates were performed, statistical significance could not be assessed. Therefore it would be more appropriate to mention the Spearman’s correlation coefficients of the two biological replicates (0.81, 0.51, 0.86 and 0.71 for the 0-4h, 4-8h, 8-12h and 12-16h samples respectively, as shown in Supplementary Figure 1). The use of Spearman’s rather than Pearson’s correlation coefficient should be justified. Scatter plots of the biological replicates would be the best way to document the reproducibility of the data.

We thank the reviewer for this comment. We fully agree with the reviewer that plotting the data is the best way to assess variability, as perhaps most prominently pointed out by Anscombe (Anscombe 1973). The only reason for not making scatter plots are space constraints. We now replaced figure S1 with a version that depicts included miniature scatter plots. We chose Spearman’s correlation coefficients instead of Pearson’s because the latter one is sensitive to outliers. Specifically, a single extreme value can lead to a high Pearson correlation coefficient, as illustrated by the fourth dataset in Anscombe’s quartet. We now also added Pearson correlation coefficients to Figure S1 for completeness.

(2) Following up with the previous comment, it should be systematically stated in the figure legends whether the data are representative of one replicate or both (the information is missing in several instances, e.g. in Figure 2, 3 and 4). Likewise, the number of biological replicates should be indicated in the legends of Figures 5c, 5d, 6b-e.

We thank the reviewer for bringing this up, it was indeed not fully clear. We now specified the number of replicates used in the figure legends.

(3) In Figure 2e, the scatter plot should be shown for both the Mal.mock and Pan.mock datasets. In Figure 2f, the exact nature of the IBAQ values (which time period) and

RPKM value (which time point) used to calculate the IBAQ / RPKM ratios should be explained. If IBAQ 8-12h and RPKM 8hpi were used, as suggested from the data shown in Figure 2d and 2e, this would correspond to a late time point in the course of infection and may not allow to conclude that viral proteins are, overall, “less efficiently synthesized than host proteins” (page 9).

We added a scatterplot of the Pan.mock infection condition to figure 2e as requested.

The iBAQ data is indeed from the 8-12h time period and the RPKM value from 8 hrs: these data showed the highest correlation, which is also expected since transcription precedes translation. We added this missing information to the legend of figure 2.

The reviewer correctly points out that this corresponds to a rather late time point. To clarify this we changed the sentence to “Instead, viral proteins were even less efficiently synthesized than host proteins in both strains during this time period.”

(4) In the discussion, the findings about the human proteome dynamics in infected A549 cells should be more thoroughly compared to and integrated with previously published proteomics studies on influenza-virus infected cells, including the authors initial comparative study using the same human Pan and avian Mal viruses as in the present study. The added value of the pulse SILAC compared to the Spike-in SILAC dataset should be better highlighted.

*To compare the steady-state data (Sadewasser et al., 2017) with our pAHA-SILAC data we generated scatter plots (**Figure R7**). On the one, there was a clear and highly significant correlation between the datasets. On the other hand, the correlation is modest, as expected due to (i) the complementarity of the methods and (ii) the overall rather minor changes in host protein levels.*

*A particular advantage of our pAHA-SILAC approach is that it can sensitively capture rather minor changes in protein synthesis upon infection. For example, we previously reported that the Pan and Mal strain induce a very modest interon response (Sadewasser et al., 2017, Fig. 1). Consistently, the previous steady state proteomic data provided little evidence for an up-regulation of interferon response genes (**Figure R7 A and B**). In contrast, pAHA-SILAC revealed a clear increase in de novo synthesis of IFN-induced proteins (**Figure R7 A and B**, see also Figure 2C in our manuscript). In this regard, it is interesting that the pAHA-SILAC for IFN-response genes correlates better with ribosome profiling data from a different study (Bercovich-Kinori et al., 2016). This further highlights the differences between measuring steady-state protein levels*

(Sadewasser et al., 2017) and protein synthesis (this study and Bercovich-Kinori et al., 2016). We therefore included the following sentence to the discussion:

“This is advantageous since it allows us to investigate changes in protein synthesis with high temporal resolution, which provides complementary information to more classical steady-state measurements”

*Since the manuscript is already quite long and these more technical points are not our main focus we decided not to include **Figure R7** in the manuscript.*

Figure R7. Cross-comparison of pAHA-SILAC data with literature data. (a,b) Scatterplot comparing fold-changes in whole cell lysates (Sadewasser et al., 2017) to changes in protein de novo synthesis, Pan and Mal as indicated. **(c,d)** Scatterplot comparing changes in ribosome protected fragments (Bercovich-Kinori et al., 2016) to changes in protein de novo synthesis. Proteins related to the Type I IFN response (GO:0060337, “type I interferon signaling pathway” are highlighted red). The Spearman rank correlation coefficients (ρ) and p -values (p) for all (black) or IFN response proteins (red) are indicated.

(5) Figure 3. The SILAC protein profile of the NS1 protein could not be determined in Figure 3a, because of a lack of shared peptides between the Pan and the Mal samples. However the authors should point out the fact that in their previous study, Mal-infected cells showed not only decreased steady-state levels of M1 protein but also increased levels of NS1 protein, compared to Pan-infected cells (Figure 1B in Sadewasser et al, Mol Cell Proteomics 2017). Therefore the production of NS1/NS2 mRNAs and proteins might well be imbalanced in Mal-infected cells, and this point should be further documented. This point is particularly relevant to the delayed nuclear export of vRNPs illustrated in Figure 3c-3d, as NS2 plays an essential role in the nuclear export of viral vRNPs. In Figure 3, it would be informative to stain the cells for M1 and NS2. Complementation experiments would be needed to demonstrate that the amount of M1 protein (or possibly, NS2 protein) is limiting.

We thank the reviewer for bringing this possible cause of misinterpretation to our attention. Importantly, figure 1B from Sadewasser et al. is a western blot that uses antibodies for NS1 detection. NS1 is the most variable protein between both strains with ~16.5 % difference at amino acid level. Since antibodies can have vastly different binding affinities to divergent antigens, fig. 1B in Sadewasser et al. does not provide convincing evidence for differences in NS1 protein amounts between both strains.

*However, we agree with the referee that it might be interesting to also assess potential differences in NS splicing. Therefore, we now also quantified NS2/NS1 splicing via splice junction reads, similar to the analysis we performed for the M segment (see **Figure R8**). This shows that overall levels of RNAs derived from the NS segment are lower for the Mal strain. Hence, there is no evidence for increased NS1 mRNA in the Mal strain. Also, the relative fraction of spliced transcripts is lower for the Mal than for the Pan strain. This indicates that NS segment splicing also differs between human- and avian-adapted strains, consistent with a paper we also cite (Huang et al., 2017). However, the differences for M segment splicing we observed are more pronounced (9-fold versus 2.6-fold). Moreover, our computational and experimental analysis indicates that differences in M segment splicing are a general feature of avian versus human-adapted strains. In contrast, NS sequence variants linked to human adaptation correlate with increased (**Figure R8**) or decreased splicing (Huang et al., 2017).*

The reviewer correctly points out that we cannot rule out that differences in NS1 protein levels do exist and may play a role for host range. This is one out of many possible factors that we did not assess in detail (posttranslational modifications of proteins, protein-protein interactions, RNA modifications, lipid composition, metabolic changes, ...). Discussing all of these possibilities without experimental evidence would be highly

speculative, which is why we focused on the strong experimental and computational evidence we have (that is, differences in M splicing). We feel that including additional analyses like the one presented in **Figure R8** distracts from the main message and would unnecessarily lengthen and complicate the manuscript. Importantly, while our manuscript reports that M segment splicing is a host range determinant, we do not claim that it is the only factor (see also our answer to reviewer #1, point 1). We stressed this throughout our manuscript, for example:

- Title: *The dynamic proteome of influenza A virus infection identifies M segment splicing as a host range determinant*
- Abstract: *Thus, our data identifies M segment RNA splicing as a viral determinant of host range.*
- Discussion: *“... it is critical to also emphasize that this is not the only relevant factor for IAV host range. For example, despite the overall similar host response, we and others have previously described host factors affecting human and avian virus infections. It is also well-established that the RdRP of avian-adapted strains is less active in human cells. Moreover, differences in the binding specificity of viral hemagglutinins (HA) are known to play an important role for host adaptation. Lastly, M-segment splicing does not only depend on cis-regulatory elements but also on trans-acting factors, such as NS1, RdRP, NS1-BP or HNRNPK. Indeed, while M1 production was clearly impaired in our mutant strain (Figure 6B), the wild-type bird-adapted strain produced even less (Figure 3A). It is therefore important to interpret our findings in the broader context of viral and host factors that jointly determine the success of IAV replication.”*

Figure R8. Comparative assessment of NS segment RNA splicing

(a) Schematic depiction of NS gene architecture. The NS segment is the second-smallest segment of IAV and encodes three major transcripts: The collinear transcript encoding NS1 (green), and two alternatively spliced transcript (NS2 mRNA) that encodes for NS2 and NS3 mRNA. The ORFs are indicated with colored boxes. In addition, the positions of the ATG codon, of splice donor and acceptor sites as well as stop codons are indicated. (b) Relative quantification of the different isoforms based on splice junction reads from RNAseq data for both strains at 8 h p i. The area of the pies reflects the absolute number of splice junction reads.

(6) The M2/M1 and NS2/NS1 mRNA ratios that can be estimated from the histograms shown in Figure 4a are surprisingly different from those published previously by several teams. In particular the NS2/NS1 ratio is much higher than the one reported upon RNAseq analysis of H3N2 influenza virus-infected cells (in the 0.5-5% range, Fabbozi et al, Journal of Virology, 2018) or upon RT-qPCR analysis of the A/WSN/33 laboratory strain (in the 10-15% range, Robb et al J Gen Virol 2009, Fournier et al PLoS Path 2014).

If RPKM for NS2 were measured at the splice junction whereas RPKM for NS1 were measured on the whole length of the mRNA, this might introduce a bias in the evaluation of the NS2/NS1 ratio because of irregular coverage. The authors should provide additional details about how RNAseq data were processed, indicate the M2/M1 and NS2/NS1 ratios as determined by RT-qPCR analysis (which is uneasy to infer from the type of graphs shown in Figure 4e), and discuss their findings with regard to the literature.

*The reviewer correctly points out that RPKM-based quantification of isoforms can be biased because of irregular coverage. This has to do with the way how Cufflinks assigns expression values to isoforms: Cufflinks uses not only junction reads but also exonic reads. This can be problematic when comparing very similar and small isoforms (as in our NS1/NS2 case) since the maximum likelihood model employed by Cufflinks does not account for differences in read coverage due to fragment sequence features (Love et al., Nature Biotechnology, 2016). **Figure R9** shows that the coverage for the second exon of NS2 is indeed higher than for the intronic sequence and the first exon. This may have numerous technical reasons, including unequal amplification by the polymerase due to differences in GC content. Also, we used poly-A enrichment which may bias the reads to the 3' end of the mRNA.*

For these reasons, the relative abundance ratios of isoforms should not be estimated based on their RPKM values. This is the reason why we exclusively used splice junction reads for the relative quantification of M segment isoforms, as stated in the results section:

“We investigated the relative proportion of these isoforms in the RNA-seq data via splice junction reads.”

*The relative proportions in the pie chart (splice junction read-based, Fig. 4 D) are not necessarily identical to the relative proportions in the bar chart (RPKM-based, Fig. 4 A). To assess the relative proportions of NS-segment splicing we performed the same splice junction read-based analysis (see **Figure R8** above). This shows that about 13 % of Pan and 5% Mal primary transcripts are spliced to NS2, which is in line with the literature the reviewer cited.*

Our qRT-PCR data is based on the relative quantification of a given variant (M1 or M2) to a loading control and a common reference point (10 h post infection). Importantly, due to differences in amplification efficiency and sensitivity a direct comparison of M1 versus M2 is not possible. The pie charts (Figure 4 D) are more informative for this

purpose (18% and 2% M2 for Mal and Pan, respectively). To avoid confusion, we therefore removed the qRT-PCR data from the manuscript.

Figure R9. Read coverage for NS segment of strain Pan and Mal in RNAseq data.

An accurate estimation of M1/M2 would require absolute quantification, which was not the point of the experiment where we used double delta Ct method, which is used for relative quantification. In particular, it is problematic to estimate the ratio of M1 to M2 from this as the polymerase chain reactions for M1 and M2 may have different sensitivities and efficiencies.

(7) The data shown in Figure 5 and Figure 6 indicate that the sequence next to the 3' splice site of the M segment can determine a higher splicing efficiency in Mal-infected than in Pan-infected cells. However they do not convincingly demonstrate that this sequence is involved in mammalian adaptation. To this end, it would have been more relevant to generate a mutant Mal virus in which the 3' splice site region of M is mutated towards the corresponding Pan sequence. In addition, the M splicing pattern upon infection with the Mal virus should have been compared in human and in avian cells.

*We tried hard to reconstitute an avian virus with a human 3' splice site using a plasmid-based pHW2000 system in HEK293T cells. Unfortunately, we could not obtain infectious progeny despite numerous attempts. This is probably due to the already low propagation efficiency of the Mal virus in mammalian cells. Therefore, we used a different strategy to assess how an avian M segment with a human 3' splice site would behave (RdRP reporter system, **figure R5** above, also see our response to reviewer #1 comment #6). This experiment shows that introducing the human-adapted splice site region into the avian M segment markedly decreases splicing. We added these data to figure 5.*

*The idea to compare infection of human and avian cells is excellent! We followed the suggestion and investigated the production of viral proteins upon infection of human (A549) and avian (DF-1) cells with the Mal strain (**Figure R10**). We infected cells and then pulse labeled with heavy or medium amino acids during 2 time intervals. We then combined the labels, lysed and looked at the A549 / DF-1 (H/M or M/H) SILAC ratio. Interestingly, this experiment revealed that M1 production is higher in avian than in human cells, suggesting that the reduced M1 production of avian viruses in human cells reflects poor adaptation to the mammalian splicing environment. We added these data to figure 4 and the following text to the results section:*

"In principle, the increased splicing of the M segment RNA by bird-adapted strains may reflect an evolutionary adaptation to different needs of the corresponding viral proteins in avian cells. In this case, avian-adapted strains would produce less M1 in both human and avian cells. Alternatively, the optimal balance of viral proteins could be constant and independent of the host species. In this case, avian-adapted strains would produce

more M1 in avian than in human cells. To investigate this experimentally, we infected both human and chicken cells with the Mal virus and monitored the production of viral proteins using pSILAC (Figure 4). We found that the Mal strain produced considerably more M1 in chicken cells than in human cells (Figure 4). Hence, the reduced M1 production of avian viruses in human cells appears to reflect poor adaptation to the mammalian splicing environment.”

Figure R10. Host-specific control over viral protein production of the Mal strain.

(a) SILAC-light labeled DF-1 and A549 cells were pulse labeled with heavy or medium heavy SILAC amino acids (in label-swap duplicates) after infection with the avian (Mal) isolate. (b) The average pSILAC fold-changes are depicted for both pulse intervals comparing the production of viral proteins with the Mal strain in human versus avian cells.

(8) It is unclear why the authors did not take into account a larger 3' splice site region than the one highlighted in figure 5e, to include the adjacent secondary structure which is predicted to occur in the human but not in the avian virus (all the more so as the latter is predicted with a higher probability than the secondary structure on which the authors chose to focus).

The referee is right. The bioinformatic approach that we performed predicts an RNA secondary structure distal (towards the 3' end) to the 3' splice site in human IAV sequences. The referee is also right that this structure has a relatively high pairing probability. Importantly, this structural element is mutually exclusive with the Hairpin depicted in Fig. 5 f.

As described in the manuscript, we first generated chimeric constructs to identify the cis-regulatory regions responsible for differential splicing. These preceded the computational analysis of conserved RNA secondary structures and identified the region 707-779 to be sufficient.

*The reviewer correctly points out that the RNA secondary structure depends on the larger entire sequence context, and the conserved secondary structure in human-adapted strains is probably also relevant. Therefore, we now included an experimental analysis of the RNA secondary structure for the entire region of interest. To this end, we cloned nt 707-825 from both strains and chimeric constructs into a vector for in vitro transcription under the control of a T3 promoter (backbone construct: Addgene 118612). The promoter region and the region of interest were then PCR amplified and used as template for in vitro transcription. We purified the RNA and ran it on native and denaturing agarose gels, which were stained with ethidium bromide (see **figure R3** above). We observed that the Pan and Mal RNA structured region has a different migration pattern in native but not denaturing agarose gels, providing evidence for different RNA secondary structure (lanes 1 and 2). Importantly, when substituting 8 nucleotides in the region 707-779 (termed 3' splice site region in our manuscript) from Mal into the Pan backbone (lane 3) we observed the same migration pattern as for the Mal wildtype RNA. Conversely, when substituting 8 nucleotides from the Pan into Mal backbone (lane 4) we observed the same migration pattern as for the Pan wildtype RNA. Thus, our conclusion that differences in secondary structure exist at the 3' splice site is supported by biochemical and bioinformatic evidence. Furthermore 8 nucleotides in the region are critical for controlling differences in secondary structure and splicing. We included this experiment in the revised version of the manuscript (Figure 5G).*

(9) To better assess the global significance of the strain-specific splicing patterns with respect to host-range, the analysis should be extended to additional human and avian viral isolates. In particular, the Mal virus used in this study might not be representative of all avian viruses, as it does not show a marked and global defect in the production of viral mRNAs and proteins at early time-points upon infection of human cultured cells, as reported in the literature for several avian influenza viruses.

Following the reviewer's comment, we assessed strain-specific splicing pattern for M segments of additional IAVs. We included A/chicken/Rostock/45/1934 (H7N1) as an early representative strain for the eastern avian lineage that was also frequently used in previous studies (see citations 3-6 in our original manuscript). This strain displayed an aggravated phenotype with even stronger M segment splicing than the Mal strain (see below **Figure R11**, lane 3). Intriguingly, introducing the human-adapted mutations at the splice site region into this construct completely reversed the splicing phenotype (lane 4). Also, the "avian-like" M segment of A/swine/Netherlands/25/1980 (H1N1) showed a splicing phenotype similar to the Mal strain. Finally, the human zoonotic strain A/Vietnam/1203/2004 (H5N1) exhibited a splicing pattern similar to the Pan strain (lane 5). Collectively, these findings support the global significance of the strain-specific splicing patterns with respect to host-range. We added these analyses to Supplementary figure S4.

Figure R11. The M segment coding sequences (nt 29-1007) of the indicated strains were cloned into the M segment splice reporter construct (pDEST26-Flag/HA). A549 cells were transfected with these constructs and M1/M2 expression was analyzed by western blotting.

Other comments.

(10) In the introduction, the information about expression of viral proteins from the M segment should be improved. The function of the M1 and M2 proteins is approximatively described. Important references about the ratio of spliced to unspliced mRNAs are missing (e.g. Robb & Fodor J Gen Virol 2012 , Tsai et al PLoS Pathogens 2013). More generally, a number of references about influenza virus biology in the introduction could be updated (e.g. Ref #7 Fodor 2013 could be replaced with the more recent te Velthuis and Fodor 2016).

We updated reference #7 as suggested.

We cited Robb & Fodor J Gen Virol 2012 as reference 66 in our original manuscript. As this reference discusses M RNA splicing and its regulation by NS1, we think that its current place in the discussion section is optimal.

We now cite Tsai et al PLoS Pathogens 2013 as reference #67 in our manuscript. As this paper discusses trans-acting factors controlling M segment RNA splicing, we feel that it fits best into the discussion section.

(11) The authors should clarify whether the M segment sequence alignments were performed on H3N2 isolates only, as stated in the Results section page 14, or on H3N2 and pH1N1 isolates, as stated in the Methods section page 37. In both cases, the rationale should be explained.

We thank the reviewer for pointing out this potential understatement. We stated in the Methods section exactly how we queried sequences from FluDB. Since we accepted only H3N2 sequences for human IAVs this excludes pH1N1 sequences. Nevertheless, the box "include pH1N1" was on the FluDB website was checked. This is a purely technical description of our analysis procedure and does not mean that pH1N1 strains were included. To avoid this misinterpretation we removed this sentence from the methods section.

The reasoning for only including H3N2 stains this is that there are currently circulating M segments in humans from two distinct adaptation events. The first occurred before 1918 and the second one by reassortment of the eurasian 'avian-like' swine with triple-reassortant swine virus. The M segment of the Pan strain that we investigated is a

product of the former adaptation event. In order to have “clean” bioinformatic data with only one of these adaptation events we excluded pH1N1 strains from the analysis.

(12) Figure 6. Panel a, it should be indicated whether the nucleotide changes result in amino acid changes.

We refer the reviewer to Supplementary Table S3, which contains this information.

(13) Analysis of M1 and M2 mRNA levels by qRT-PCR. Was the M1/M2 specificity assessed ?

Blasting the primer sequences resulted in no target in Homo sapiens RefSeq mRNAs. To test the M1/M2 specificity we performed dilution experiments (Figure R12) and observed high specificity and sensitivity for M1 (specificity 99.99%; efficiency 92.24%) and M2 (specificity 99.97%; efficiency 99.99%). Nevertheless, in response to point 6 above, we removed the qRT-PCR data from the revised manuscript and replaced it with RNAseq data to avoid confusion.

Figure R12. qRT-PCR standard curves of M1/M2. Serial dilutions of known M1 and M2 molar amounts (obtained from restricted plasmid DNA) were subjected to PCR amplification using M1 or M2 primers in triplicates. The obtained standard curves were

used to estimate PCR efficiency (slope). M1/M2 specificity was assessed on the basis of Cts from off-target and target amplifications.

(14) Page 21, the statement « we are living in a pandemic era of IAV infections that began at around 1918 » is misleading, as it may suggest that there have been no influenza pandemics before 1918.

This statement is based on the given reference (Morens et al., 2009). The relevant section in this paper reads “A useful way to think about influenza A events of the past 91 years is to recognize that we are living in a pandemic era that began around 1918.” We think this is indeed a useful thought, which is why we mentioned it. Our next sentence is “At this time, a virus of ultimately avian origin acquired the ability to spread among humans and later on contributed its genetic material to other pandemic viruses until present.” We think this makes the context of the statement clear. We do not imply that there have been no other influenza pandemics before 1918.

Reviewers' Comments:

Reviewer #1:

Remarks to the Author:

The authors have addressed all of my concerns in this version of the manuscript

Reviewer #3:

Remarks to the Author:

In the new version of their manuscript Bogdanow et al have thoroughly addressed the concerns raised by the reviewers.

To follow up with my comment #6 : given the large overlap between the spliced and unspliced viral mRNAs, I would recommend to add information in the legend of Figure 4a on how RNAseq reads were assigned to M1 versus M2 mRNAs and NS1 versus NS2 mRNAs. This will help readers understand the difference between the data shown in Figure 4a and 4d.

Point by point response to the reviewer

Reviewer #3 (Remarks to the Author):

In the new version of their manuscript Bogdanow et al have thoroughly addressed the concerns raised by the reviewers.

To follow up with my comment #6 : given the large overlap between the spliced and unspliced viral mRNAs, I would recommend to add information in the legend of Figure 4a on how RNAseq reads were assigned to M1 versus M2 mRNAs and NS1 versus NS2 mRNAs. This will help readers understand the difference between the data shown in Figure 4a and 4d.

We added this information to the legend of Figure 4 as suggested.